# Tetrahedral DNA dendritic nanostructure-enhanced FISH for high-speed, sensitive spatial transcriptomics

Yi-Fan Wang[1], Hua-Jie Chen[1], Zhong-Da He[1], Zhi-Gang Wang ®[2], Dai-Wen Pang ®[1] & Shu-Lin Liu ®[1]✉

Understanding where genes are active within tissues is essential to explain how cells build and maintain organs, yet many spatial RNA assays are slow and weak, limiting short-transcript detection and masking cellular diversity. Here we show that TDDN-FISH (Tetrahedral DNA Dendritic Nanostructure–Enhanced Fluorescence In Situ Hybridization), a rapid, enzyme-free method using self-assembling DNA nanostructures, accelerates and amplifies RNA detection. Per round, TDDN-FISH is ~eightfold faster than HCR-FISH and generates stronger signals than smFISH, enabling short-RNA detection and low-magnification tissue imaging with single-cell and subcellular resolution. Iterative, multiplexed hybridization produces color-coded readouts for many targets in the same specimen, supporting high-throughput spatial transcriptomics. We apply TDDN-FISH to cultured cells and tissue sections to map RNA distributions with high specificity and reveal complex expression patterns. This platform streamlines workflows and broadens access to spatial RNA profiling for studies of cellular heterogeneity, tissue organization, and disease mechanisms.

Understanding gene expression in its spatial context is essential for unraveling the functional complexities of multicellular structures[1,2]. Spatially resolved gene expression profiling identifies distinct cell subpopulations and their interaction networks, offering new insights into tissue development, differentiation, and morphogenesis[3,4]. Fluorescence in situ hybridization (FISH) technique provides high-sensitivity detection of RNA at single-molecule resolution, enabling simultaneous analysis of multiple genes in individual cells[5-7]. However, its low signal output and fluorophore spectral overlap limit the number of detectable RNA species, complicating studies of cellular heterogeneity in complex tissues. There is an urgent need for advanced spatial transcriptomics techniques capable of detecting more RNA targets with single-molecule precision, essential for understanding complex tissue environments and cell interactions, and advancing developmental biology, tissue engineering, and precision medicine.

The emergence of combinatorial coding strategies for in situ hybridization (e.g., MERFISH, seqFISH+, Split-FISH) has transformed the landscape of spatial transcriptomics, enabling unprecedented multiplexing capabilities for detecting hundreds to thousands of RNA species simultaneously[8-10]. Despite these remarkable advances, several critical challenges persist, limiting their widespread adoption and application. A fundamental constraint lies in the requirement for long RNA transcripts (>1.5 kb) to achieve optimal signal-to-noise ratios, effectively excluding the analysis of biologically important short RNA species[11]. Furthermore, the accumulation of background noise from multiplexed probe hybridization poses significant challenges in signal discrimination, particularly in densely packed or suboptimal tissue samples. While padlock probes with rolling circle amplification (RCA) offer high sensitivity, specificity, and multiplexing for spatial transcriptomics imaging, making them powerful for studying gene expression in cells and tissues[11-13], these methods are typically costly

[1]State Key Laboratory of Medicinal Chemical Biology, Frontiers Science Centre for New Organic Matter, Tianjin Key Laboratory of Biosensing and Molecular Recognition, Research Centre for Analytical Sciences, Frontiers Science Center for Cell Responses, College of Chemistry, Nankai University, Tianjin, PR China. [2]School of Medicine, Nankai University, Tianjin, PR China. ✉e-mail: shulin.liu@nankai.edu.cn

and time-consuming, requiring long amplification periods. Moreover, variability in enzymatic activity due to differences in cellular environments further complicates the consistency of imaging signals. Variability in enzyme activity due to differences in cellular environments further complicates imaging signal consistency. Other amplification strategies, such as tyrosine signal amplification[14] and enzyme-mediated fluorescence[15], while promising, continue to face limitations in achieving consistent and efficient signal generation across complex tissue architectures. These technical bottlenecks highlight an urgent need for next-generation spatial transcriptomics platforms that integrate improved speed, sensitivity, and throughput.

Tetrahedral DNA nanostructures have emerged as a paradigm-shifting tool in molecular biosensing, redefining the boundaries of analytical performance through their programmable architecture and physicochemical robustness[16–19]. These Nanostructures exhibit exceptional thermal and chemical stability, retaining precise spatial organization even in complex biological matrices—a critical advantage for in situ applications[20,21]. The capacity for modular self-assembly enables precise engineering of functional domains, including multivalent probe conjugation sites and enzymatic amplification modules, while maintaining nanoscale spatial control[16]. This programmability allows systematic optimization of probe density, hybridization kinetics, and steric accessibility—parameters crucial for resolving spatially constrained RNA targets. Furthermore, their three-dimensional framework supports hierarchical assembly of dendritic signal amplification units, achieving exponential enhancement of fluorescence output without compromising target specificity—addressing the sensitivity-throughput trade-off in conventional FISH methodologies.

Capitalizing on these foundational advances, we developed TDDN-FISH (Tetrahedral DNA Dendritic Nanostructure-Enhanced Multiplexed FISH), an improved spatial omics platform integrating three key technologies: (1) Cascade Self-Assembly Protocol for precise formation of hierarchical TDDNs, enhancing detection sensitivity and specificity; (2) Cyclic Encoding-Decoding Framework enabling simultaneous detection of $F^N$ RNA types via multiplexed fluorescence (F) and iterative hybridization (N) and (3) Reference scRNA-seq Data for simultaneous labeling of multiple mRNAs in a single imaging session, reducing imaging time and complexity while maintaining high resolution (Fig. 1). Benchmarking against traditional methods such as HCR-FISH and smFISH demonstrated that TDDN-FISH achieves improved performance metrics, including eightfold faster single-round detection compared to HCR-FISH and significantly stronger signal intensity than smFISH, enabling reliable detection of low-abundance RNAs, including short RNAs such as miRNAs. A key advantage of this platform is its enzyme-free structure, which eliminates batch variability caused by fluctuations in DNA ligase or polymerase activity, ensuring the robustness and reproducibility of experimental results. Additionally, the customized dendritic probes of this platform enable subcellular resolution in complex samples such as cells and tissues, providing nanoscale, detailed spatial information. This platform provides enhanced sensitivity and speed, making it well-suited for analyzing complex gene expression patterns across various biological systems.

## Results

### Establishment of TDDN-FISH method

To significantly enhance signal amplification in FISH experiments, we engineered a Tetrahedral DNA Dendritic Nanostructure (TDDN) through a precisely controlled layer-by-layer self-assembly strategy (Fig. 1a). The design incorporates three distinct tetrahedral DNA monomers (T0, T1, and T2), each with a side length of 17 base pairs and a calculated hydrodynamic diameter of 5.8 nm, assembled from four complementary oligonucleotide strands (Fig. 2a)[16,22]. Each monomer is functionalized with four single-stranded overhangs, serving as programmable recognition sites for hierarchical assembly. The T0 monomer forms the structural core of the TDDN, featuring four

strategically designed sticky ends: one for conjugation with the primary RNA-targeting probe and three for initiating the first layer of dendritic growth (Shell-0). The T1 monomer, constituting the first dendritic layer (Shell-1), is engineered with four sticky ends—one complementary to T0 and three complementary to T2. The T2 monomer, forming the second dendritic layer (Shell-2), is similarly designed with four sticky ends: one complementary to T1 and three for coupling with fluorophore-labeled oligonucleotide strands. This modular, layer-by-layer assembly (Shell-0 + Shell-1 + Shell-2) yields a highly branched DNA Nanostructure with exponential signal amplification capacity[23]. The successful assembly of the T0, T1, and T2 tetrahedra was rigorously validated using 2% agarose gel electrophoresis, which revealed distinct bands corresponding to each monomeric unit and their intermediate assembly products (Supplementary Fig. 1a). Furthermore, atomic force microscopy (AFM) imaging provided direct visualization of the layered DNA Nanostructures, confirming their structural integrity and nanoscale dimensions (Fig. 2b and Supplementary Fig. 1b). These results collectively demonstrated the robust and reproducible construction of the TDDN, establishing a foundation for their application in high-performance FISH imaging.

Next, we leveraged the TDDN architecture as a high-performance FISH probe to detect the highly expressed *ACTB* mRNA in HeLa cells. The experimental workflow involved the hybridization of a bifunctional primary probe—comprising a target-specific sequence for binding endogenous *ACTB* mRNA and a readout sequence for subsequent TDDN attachment—to the cellular mRNA targets. The TDDN, functionalized with complementary sticky ends, were then introduced to specifically bind the readout sequence of the primary probe, enabling robust TDDN-FISH imaging analysis (Fig. 2c). Confocal imaging demonstrated a stepwise enhancement in fluorescence intensity corresponding to the sequential assembly of the TDDN layers, culminating in a significantly improved fluorescence intensity for *ACTB* mRNA detection (Fig. 2d). Control experiments confirmed that the strong RNA FISH signal was contingent on the presence of all three components: the primary probe, TDDN, and fluorophore-labeled single-stranded DNA (Supplementary Fig. 1c), validating the specificity and modularity of the system. To maximize the performance of TDDN-FISH, we systematically optimized key hybridization parameters, including incubation temperature (37–42 °C) and formamide concentration (ranging from 10 to 30%), to achieve optimal probe binding efficiency and signal intensity (Supplementary Fig. 1d–g). These optimizations ensured high specificity and minimal background noise, further enhancing the reliability of mRNA visualization.

To rigorously evaluate the imaging speed and sensitivity of TDDNs as FISH probes, we performed comparative confocal imaging of cells labeled with TDDN-FISH alongside two established methods[6,24]: smFISH and HCR-FISH (Fig. 2e, f). Fluorescence imaging revealed that TDDN-FISH—using only 3 primary probes—achieved significantly higher signal intensity than both smFISH and HCR-FISH, owing to its dendritic amplification architecture for exponential signal multiplication. Crucially, TDDN-FISH enabled dramatically faster processing: each imaging round required just ~1 h post-hybridization, compared to ≥8 h for HCR-FISH amplification. While smFISH also operates within ~1 h, it relies on 48 primary probes and is constrained by target mRNA length. Additionally, to further validate TDDN-FISH's capability for short RNA detection, we successfully imaged *miR-21* (72 nucleotides in length) using just a single primary probe (Supplementary Fig. 2a, b). This demonstrates TDDN-FISH's effectiveness for visualizing small RNAs with minimal probe requirements. This combination of rapid imaging, high sensitivity, and minimal probe requirements underscores TDDN-FISH's superiority for high-throughput applications.

To further validate the specificity of TDDN-FISH, we employed a dual-fluorescence labeling strategy to visualize mCherry-tagged mRNA in cells. Specifically, we co-transfected cells with two

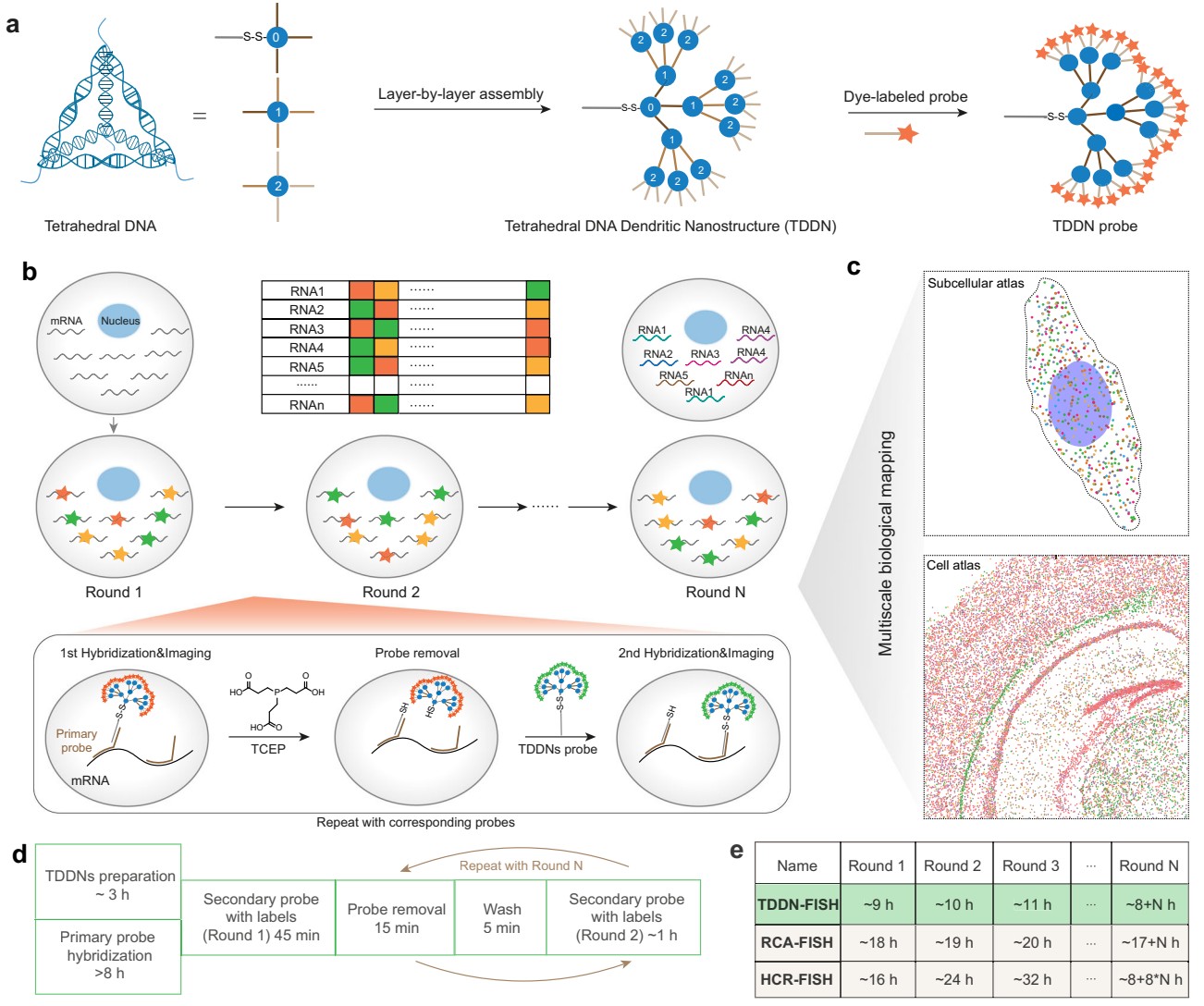

**Fig. 1 | Schematic diagram of probe design and spatial transcriptome image analysis for TDDN-FISH. a** Illustration of the assemble of Tetrahedral DNA dendritic Nanostructure (TDDN). **b**, **c** Illustration of the principle of spatial imaging transcriptomics based TDDN-enhanced FISH (TDDN-FISH). Multiple rounds of TDDN-FISH are used to color-code different RNAs, allowing for the visualization of their spatial distribution. Each round of TDDN-FISH includes probe hybridization, imaging, and probe removal with Tris (2-carboxyethyl) phosphine hydrochloride (TCEP hydrochloride). **d** Procedures and timing for spatial imaging transcriptomics by TDDN-FISH. **e** Comparisons of processing time between TDDN-FISH and existing RCA-FISH and HCR-FISH.

constructs: (1) an mRNA encoding the SARS-CoV-2 envelope (E) protein, engineered with a 24-copy PP7 hairpin sequence, and (2) a plasmid expressing PCP fused to three copies of the mCherry for fluorescent labeling of individual mRNA molecules (Fig. 2g)[25]. Following cell fixation, we performed TDDN-FISH using a green fluorescent TDDN probe targeting the mCherry-labeling mRNA. Dual-channel confocal imaging revealed >90% co-localization between TDDN-FISH signals and mCherry fluorescence (Fig. 2h, i), demonstrating exceptional target specificity of TDDN-FISH. To evaluate TDDN-FISH performance across diverse biological samples, we analyzed *POLR2A* and *HSP70* in HeLa cells, along with *Satb2* and *Gad1* in mouse brain cryosections. Comparative analysis with smFISH revealed TDDN-FISH maintained consistently low false-positive and false-negative rates in both cell and tissue samples, achieving >90% detection accuracy (Supplementary Fig. 2c–e).

## Verification of universality and efficiency of TDDN-FISH technique

To rigorously assess the universality and efficiency of the TDDN-FISH technique for imaging diverse RNA species within cells, we selected four representative RNA molecules—*mTOR* mRNA, *B2M* mRNA, *MALAT1* lncRNA, and *NEAT1* lncRNA—each exhibiting distinct subcellular localization patterns and expression levels. These RNAs were chosen to evaluate the technique's ability to resolve spatial and quantitative differences across a range of RNA classes and cellular compartments. Confocal imaging revealed striking differences in the spatial distribution of these RNAs (Fig. 3a–c). *B2M* mRNA and *MALAT1* lncRNA were detected in both the nucleus and cytoplasm. Notably, *MALAT1 lncRNA* exhibits higher expression levels in the nucleus compared to the cytoplasm, in line with prior studies[26,27]. In contrast, *mTOR* mRNA was predominantly localized in the cytoplasm, aligning with its function in cytoplasmic signaling pathways, while *NEAT1* lncRNA was exclusively nuclear, consistent with its role in nuclear gene regulation (Fig. 2b)[28,29]. Quantitative analysis of RNA spot counts further validated the technique's ability to discriminate between RNA species based on expression levels. The spot density for *B2M* mRNA was significantly higher than that of *MALAT1* lncRNA, while *mTOR* mRNA and *NEAT1* lncRNA exhibited intermediate and low spot densities, respectively. These results not only confirmed the sensitivity of TDDN-FISH for detecting RNAs across a wide dynamic range of expression levels but

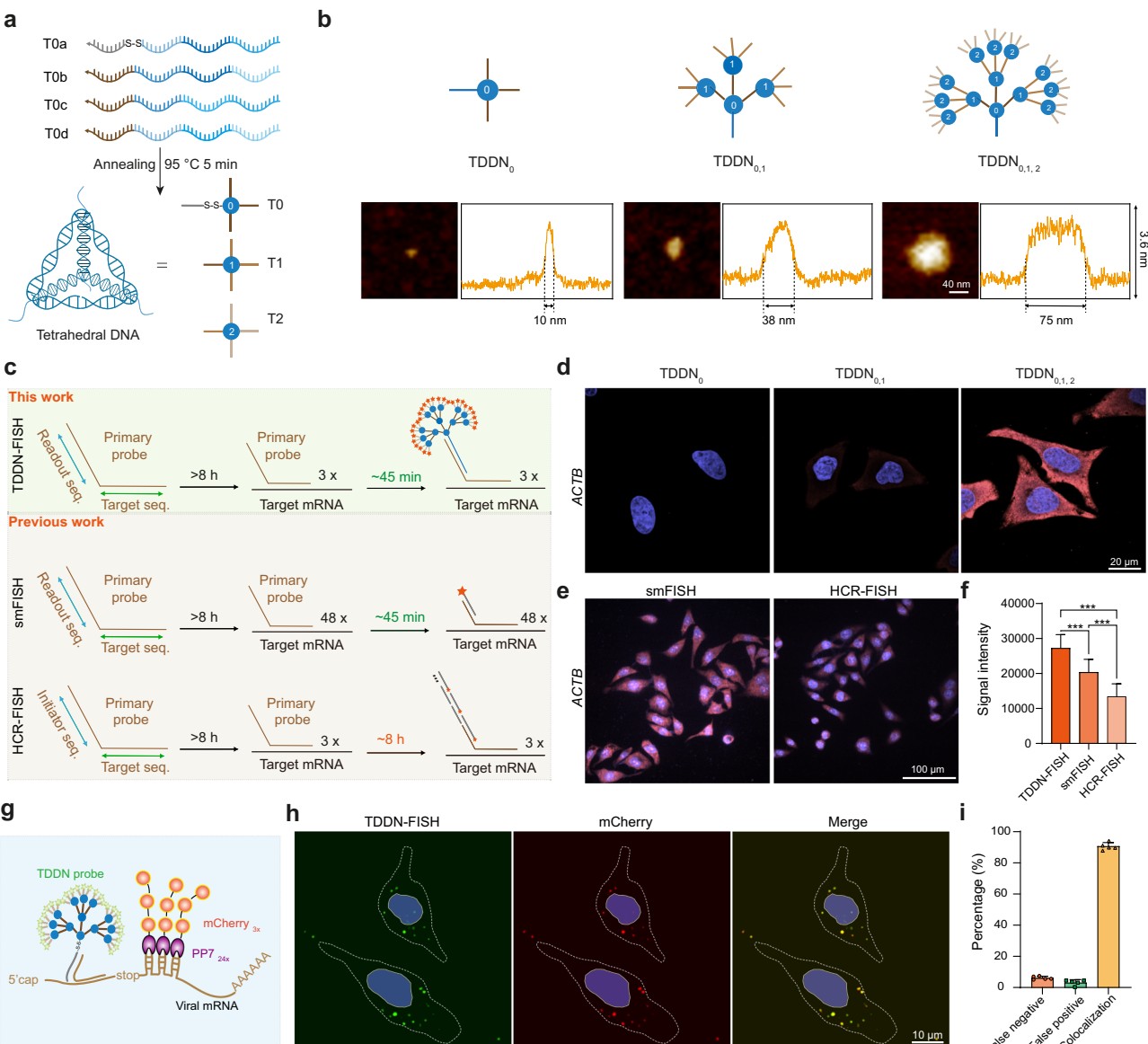

**Fig. 2 | Highly efficient and rapid TDDN-FISH for in situ detection of RNAs within cells. a** Schematic of the synthesis of tetrahedral DNAs as the TDDN monomers. Each TDDN probe contains three layers of DNA tetrahedra, called T0, T1, and T2. Each tetrahedral DNA consists of four single-stranded DNAs (e.g., T0a, T0b, T0c, and T0d). **b** Schematic structures of TDDN$_0$, TDDN$_{0,1}$, TDDN$_{0,1,2}$ with representative atomic force microscopic (AFM) images and line profiles across both terminals of the image. The full width at half maximum height (FWHM) is measured from the line profiles. Scale bars, 40 nm. **c–f** Comparison of signal intensities of TDDN-FISH, smFISH, and HCR-FISH by detecting *ACTB* mRNA in HeLa cells. **c**. Schematic representation of the FISH methodology. **d** Confocal images of *ACTB* mRNA signal generated by FISH using TDDN$_0$, TDDN$_{0,1}$, TDDN$_{0,1,2}$ as probes. **e** Confocal images of *ACTB* mRNA signal generated by using smFISH and HCR-FISH. Scale bars, 100 µm (*n* = 30, biological replicates). **f** Quantification of fluorescence intensity of cells labeled with TDDN-FISH, smFISH and HCR-FISH. **g**, **h** Analysis of the ability of FAM-labeled TDDN-FISH probes to target mCherry-labeled single mRNAs. Scale bar, 10 µm. **i** Quantitative analysis of the colocalization signals (yellow), false positive signals (green), and false negative signals (red) shown in (**h**) (*n* = 5, biological replicates). In (**f**) and (**i**), the data are presented as mean ± s.d. (ns not significant, *P* > 0.05; **P* < 0.05; ***P* < 0.01; ****P* < 0.001; unpaired two-tailed Student's *t* test). Source data are provided as a Source Data file.

also highlighted its capability to resolve subcellular localization patterns with high precision.

High-resolution RNA imaging analysis is critical for elucidating spatial gene expression patterns in both individual cells and complex tissue samples, necessitating high-brightness single-molecule labeling techniques. To further validate the versatility and robustness of the TDDN-FISH technique in complex biological systems, we applied it to frozen brain tissue sections from mice, successfully detecting five mRNA molecules with distinct expression levels and functional characteristics: *Satb2*, *Gad1*, *Slc17a6*, *Lmnb1*, and *Slc17a7* (Fig. 3d). Confocal imaging revealed highly specific spatial distribution

patterns for each mRNA within the tissue sections. For instance, *Satb2* and *Gad1* mRNAs, while their expression levels are much lower compared to housekeeping genes such as *ACTB*, were still clearly visualized in the nucleus and peri-nuclear regions, highlighting the technique's sensitivity for detecting low-abundance transcripts[30]. In contrast, *Lmnb1* mRNA was exclusively localized within the cell nucleus, consistent with its role in nuclear envelope integrity. Furthermore, *Slc17a7* mRNA exhibited a distinct spatial expression pattern, reflecting its cell-type-specific localization within neuronal populations. These observations underscore the ability of TDDN-FISH to resolve fine-grained spatial and quantitative differences in

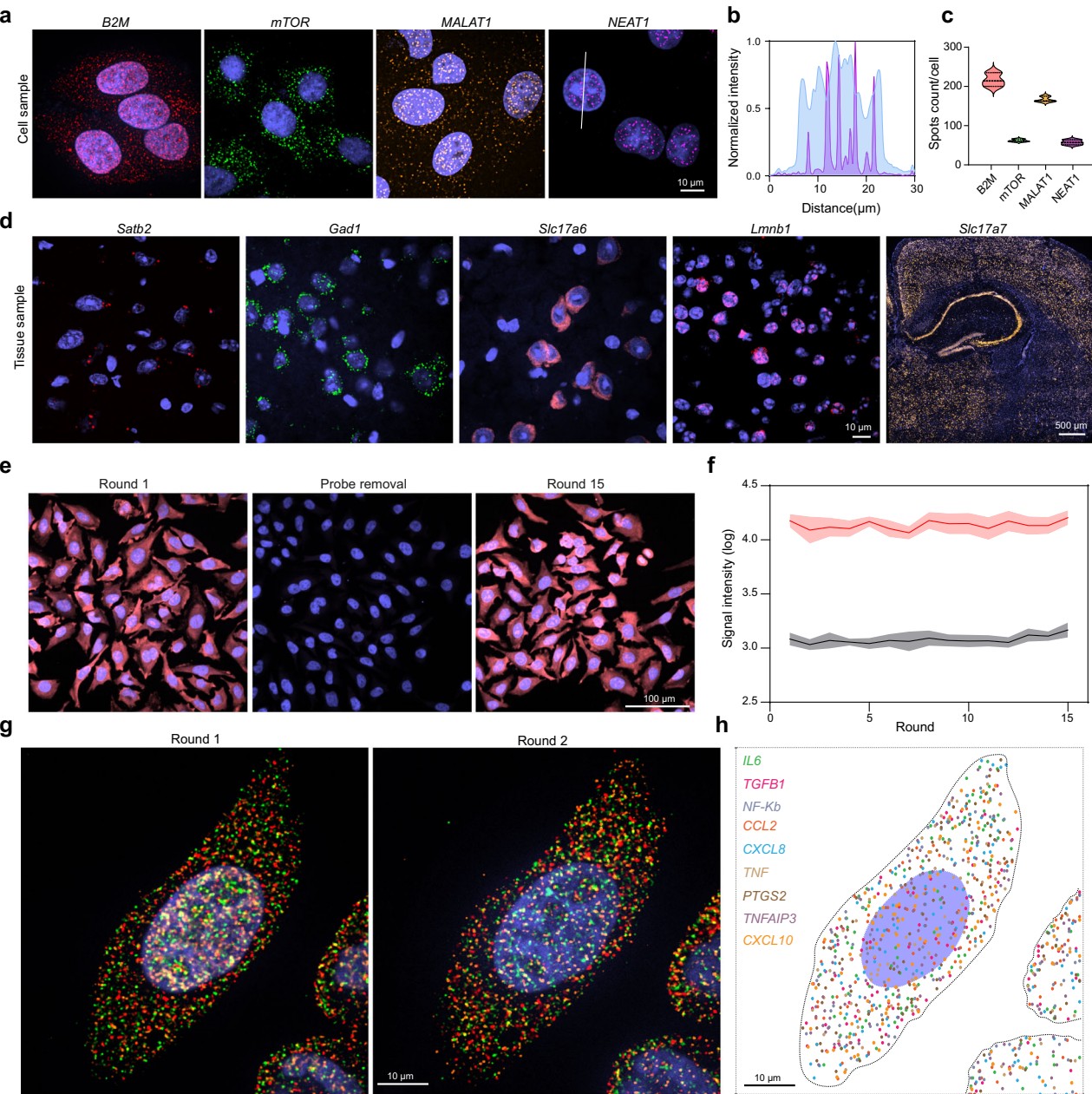

**Fig. 3 | Extended application of TDDN-FISH for the detection of different mRNAs. a** TDDN-FISH used to detect *B2M* mRNA, *mTOR* mRNA, *MALAT1* lncRNA, and *NEAT1* lncRNA in HeLa cells. Scale bar, 10 µm. **b** The line profile along the white line in (**a**) confirming that *NEAT1* lncRNA is mainly localized in the nuclear region. The blue line denotes the nuclear signal, and the purple line indicates the TDDN-FISH labeled *NEAT1* lncRNA signal. **c** Histogram showing the numbers of each RNA spot per cell (*n* = 3). **d** TDDN-FISH applied to detect *Satb2* mRNA, *Gad1* mRNA, *Slc17a6* mRNA, *Lmnb1,* and *Slc17a7* mRNA in frozen tissue sections. Scale bar, 10 µm and 500 µm. **e** Fluorescence images over 15 rounds of TDDN-FISH. Scale bar, 100 µm. **f** Lines indicating the mean fluorescence intensity of cells (*n* = 15) after hybridization (red line) and cleavage (black line), respectively. The shadow of each line indicates the standard deviation. **g, h** The mRNAs for nine cytokine or chemokine genes in HeLa cells were simultaneously detected by TDDN-FISH. Scale bar, 10 µm. Source data are provided as a Source Data file.

RNA localization within complex tissue architectures. These results conclusively demonstrated that the TDDN-FISH technique is capable of robustly amplifying and detecting RNA signals across a wide range of sample types, from cultured cells to intricate tissue sections. To assess TDDN-FISH performance in thick tissue specimens, we successfully detected *Gad1* mRNA distribution in 40 µm mouse brain sections. The method generated robust and uniform signals throughout the tissue depth (Supplementary Fig. 3a), demonstrating its effectiveness for volumetric RNA analysis.

To enhance the throughput of RNA detection based on accurate recognition and sensitive labeling of target RNA, we further developed a TDDN-based optical encoding strategy and multiround imaging method. By creating a unique barcode for each target RNA molecule in N rounds of FISH imaging, TDDN-FISH provides scalable encoding capability using an $F^N$ (F-color, N-round) approach[31,32]. Furthermore, to sequentially label target RNA populations with N-bit barcodes, we employed TDDN labeling combined with TCEP-induced disulfide bond cleavage to achieve rapid probe hybridization and dissociation for iterative RNA labeling imaging[33]. Using single-particle localization algorithms, we localized individual RNA molecules and identified their corresponding barcodes based on the color coding observed in multiple rounds of imaging (Fig. 1b).

We performed multiple rounds of FISH imaging to evaluate the feasibility of iterative labeling of *PPIA* mRNA on HeLa cells using TDDN probes and TCEP cleavage (Fig. 3e). Fluorescence images showed that hybridization probes could be effectively removed by the TCEP buffer, leading to a significant reduction in fluorescence intensity. After 15 rounds of hybridization and dissociation, the average fluorescence intensity of RNA in the cells remained approximately 10 times higher than the background (Fig. 3f). This indicates that the TCEP-based probe removal method has minimal impact on subsequent hybridization rounds, supporting the feasibility of continuous multi-round FISH imaging in cells. To further reduce experimental duration, we optimized the hybridization time for primary probes, the hybridization time for TDDN probes, and the TCEP cleavage time (Supplementary Fig. 3b–d). Ultimately, we selected 8 h for primary probe hybridization, 45 min for TDDN probe hybridization, and 15 min for TCEP cleavage.

To further validate the feasibility of our method in the application of subcellular imaging spatial transcriptomics, we performed TDDN-FISH detection on nine endogenous genes in HeLa cells. In each imaging round, three-color fluorescent probes were used to encode and label the target RNA molecules (Fig. 3g). Subsequently, we assigned a unique barcode to each RNA molecule through two rounds of image decoding. Our color-coded analysis demonstrated the spatial distribution of these nine RNA molecules within the cell (Fig. 3h). These results indicated that this technology not only enables us to analyze the precise localization of RNA within the cell but also holds promise for providing important insights into the interactions between RNA molecules for further research.

## Mapping subcellular atlas of influenza A virus RNA using TDDN-FISH

The influenza A virus (IAV) genome comprises eight RNA segments that encode essential proteins for viral function[34]. To understand the mechanisms of genome release, replication, and nuclear export during infection, it is vital to visualize these vRNAs simultaneously in cells. Although current imaging techniques demonstrate sensitivity in detecting individual vRNA molecules[35,36], methods for simultaneous imaging of the eight vRNA of IAV are still lacking. To evaluate the broader application potential of the TDDN-FISH technique in cells, we conducted single-molecule imaging analyzes of eight vRNA segments within IAV-infected cells. Initially, we designed two types of TDDN probes to label the vRNA segments encoding the *PA* and *PB1* proteins. Through colocalization analysis at various time points post-infection (hpi), we found that at 1 hpi, *PA* and *PB1* vRNA molecules predominantly localized near the cell membrane, showing nearly complete colocalization (Fig. 4a–c). This indicated that most vRNA had not yet translocated from the cytoplasm to the nucleus, remaining closely associated within the cytoplasm. By 8 hpi, the quantities of *PA* and *PB1* vRNA significantly increased, distributing almost uniformly throughout the cytoplasm and nucleus, and the degree of colocalization between the two vRNAs decreased significantly, suggesting that vRNA had entered the nucleus for extensive replication and subsequently dispersed widely throughout the cytoplasm at 8 hpi (Fig. 4a–c).

We employed the TDDN-FISH technology to color-code and analyze the vRNAs of IAV through multiple rounds of imaging. By constructing high-resolution spatiotemporal maps of vRNA at various time points during IAV infection, we clearly depicted the dynamic distribution patterns of vRNA within the host cells and their corresponding molecular events (Fig. 4d–f). At 1 h post-infection (hpi), all eight vRNAs were predominantly localized in the cytoplasm in a co-localized manner. By 4 hpi, they concentrated predominantly in the perinuclear region while maintaining high co-localization. At 5 hpi, numerous discrete vRNAs appeared in the nucleus, potentially marking the onset of viral RNA replication. Subsequently, during the period from 6 to 8 hpi, various vRNAs exported from the nucleus accumulated

extensively in the cytoplasm, suggesting that the assembly and release of viral particles were in preparation. In conjunction with literature reports, Fig. 4g illustrates the complete dynamic process of IAV from entry into host cells to the replication and nuclear export of various vRNAs. These results provide crucial insights into the infection cycle of influenza virus and its molecular interactions with the host, while also validating the powerful capabilities and broad applicability of our TDDN-FISH method in the study of viral infections.

## Mapping cellular neighborhoods in the mouse brain

To validate the potential application of TDDN-FISH technology at the tissue level, we employed TDDN-FISH on mouse brain slices to reveal single-cell spatial transcriptomics within intact tissue. To achieve a cell atlas with single-cell resolution in the tissue, we first selected six cell types from the brain: glutamatergic neurons, GABAergic neurons, astrocytes, microglia, oligodendrocytes, and endothelial cells (Fig. 5a). Based on previously reported single-cell RNA-seq datasets of mouse brain cell types, we identified the top 10 marker genes for each cell type (Fig. 5b and Supplementary Figs. 4, 5)[37]. Subsequently, we designed corresponding TDDN-FISH probes targeting a total of 60 target genes across these six cell types. During the probe design process, we cleverly applied an optical coding strategy to distinguish multiple cell types (Fig. 5c). Specifically, we utilized the same optical code for the 10 marker genes within each cell type, allowing us to differentiate cell types in the tissue based on the overall optical coding differences presented by each cell type. Following this, we assigned unique color barcodes to each cell type through multiple rounds of imaging, enabling precise differentiation and spatial localization of various cell types (Supplementary Fig. 6).

According to the procedure shown in Fig. 5d, we obtained a cellular atlas displaying distinct spatial patterns across six cell types. Glutamatergic neurons are primarily distributed in the cortex, hippocampus, thalamus, and midbrain, with the highest abundance found in the cortex, thalamus, and midbrain. GABAergic neurons, microglia, and endothelial cells exhibit similar distribution patterns to glutamatergic neurons, albeit at lower abundances. Astrocytes are found in higher abundance within the hippocampus, while oligodendrocytes are primarily located near the external capsule, with sparse distributions in other regions (Fig. 5e, f). In the magnified view of Region 1 in Fig. 5g, the cortex shows a predominance of glutamatergic neurons, with other cell types present at lower levels. Region 2 highlights the hippocampus, demonstrating that its structure is largely composed of glutamatergic neurons, with a higher concentration of astrocytes in this area. Furthermore, the technology has achieved a comparable distribution of cell types to those detected by previously reported ISS and osmFISH techniques in rapid multi-gene analyses[7,38]. These results indicated that TDDN-FISH can efficiently map cellular atlases at single-cell resolution in fresh frozen tissue samples, providing robust technical support for in-depth studies of multicellular systems.

## Visualization of the organization of cell types in the mouse primary visual cortex

Next, we applied TDDN-FISH to distinguish neuronal subtypes within the classical layered structure of the visual cortex. Excitatory neurons exhibit distinct molecular characteristics and spatial organization according to cortical layering rules, reflecting the functional differentiation of neurons at different cortical levels in visual information processing. We initially selected nine cell types: excitatory neurons (L2/3, L4, L5, and L6), inhibitory neurons (Pvalb, Sst, Vip, and Npy), and oligodendrocytes[39]. Based on previously reported scRNA-seq datasets of the mouse visual cortex, we identified the top 10 marker genes for each cell type (Fig. 6a–c and Supplementary Figs. 7, 8)[40]. Subsequently, we designed TDDN-FISH probes for a total of 90 target genes corresponding to the aforementioned nine cell types, with the target genes for oligodendrocytes being the same as those used in probes for

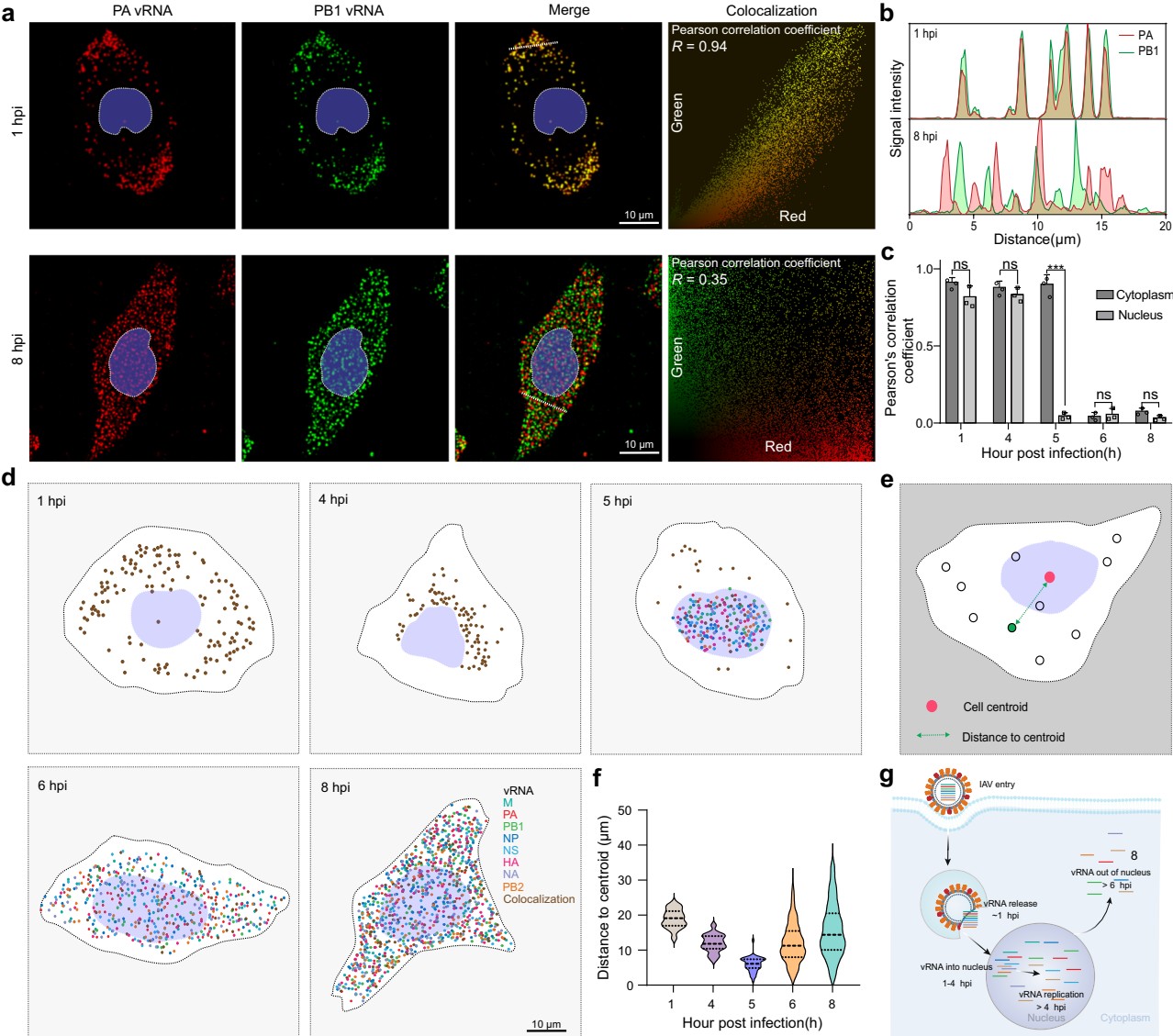

**Fig. 4 | Temporal and spatial mapping of RNAs of influenza virus using TDDN-FISH. a** TDDN-FISH imaging of *PA* vRNA and *PB1* vRNA at 1 h and 8 h post influenza virus infection. **b** Line profiles indicated by white lines are shown in (**a**).
**c** Colocalization analysis of *PA* vRNA and *PB1* vRNA in the nucleus and cytoplasm at different time points post influenza virus infection (*n* = 3, biological replicates). **d** Temporal and spatial mapping of influenza virus infection through two rounds of

imaging. Scale bar, 10 μm. **e** Schematic diagram of spot distance statistics.
**f** Statistical analysis of vRNA distance from the nucleus at different time points post influenza virus infection. **g** Schematic diagram of the mechanism of influenza virus infection. In **c**, the data are presented as mean ± s.d. (ns not significant, *P* > 0.05; *P* < 0.05; **P* < 0.01; ***P* < 0.001; unpaired two-tailed Student's *t* test). Source data are provided as a Source Data file.

mouse brain imaging. Through two rounds of imaging and fluorescent signal decoding, we explored the gene expression characteristics and spatial distribution patterns of different cell types within the primary visual cortex (V1) of mice (Fig. 6a).

Through TDDN-FISH, we obtained spatial distribution maps of different cell types in the V1 region of the mouse visual cortex (Fig. 6d–i). Excitatory neurons (L2/3, L4, L5, L6) exhibited significant spatial organization along the cortical layers, while inhibitory neurons (Pvalb, Sst, Vip, Npy) showed a more uniform distribution across layers, reflecting specific spatial distribution patterns of different cell types in the V1 region (Fig. 6d, e). Further quantitative validation of these spatial distribution patterns was performed using Kernel Density Estimation (Fig. 6f, g). Figure 6f shows the normalized distance distribution of excitatory neurons from the cortical edge across layers, indicating a clear laminar organization for L2/3, L4, L5, and L6 neurons that is highly consistent with the known anatomical

structure of the visual cortex (Fig. 6f–h). In contrast, the inhibitory neurons Pvalb, Sst, Vip, and Npy exhibited a more uniform distribution throughout the cortex, supporting their broad distribution function in regulating local network activity (Fig. 6g–i). In summary, these results suggested that TDDN-FISH can be widely applied to the construction of spatial maps of cells guided by single-cell RNA-seq data.

## Discussion

Deciphering the spatial regulation of gene expression is essential for unraveling the functional architecture of multicellular systems[1,2,41]. However, current methods are hindered by limitations such as inadequate signal amplification and time-consuming workflows, which restrict their ability to resolve cellular heterogeneity within complex tissue microenvironments. To address these challenges, we present TDDN-FISH (TDDN-Enhanced Fluorescence In Situ Hybridization), a

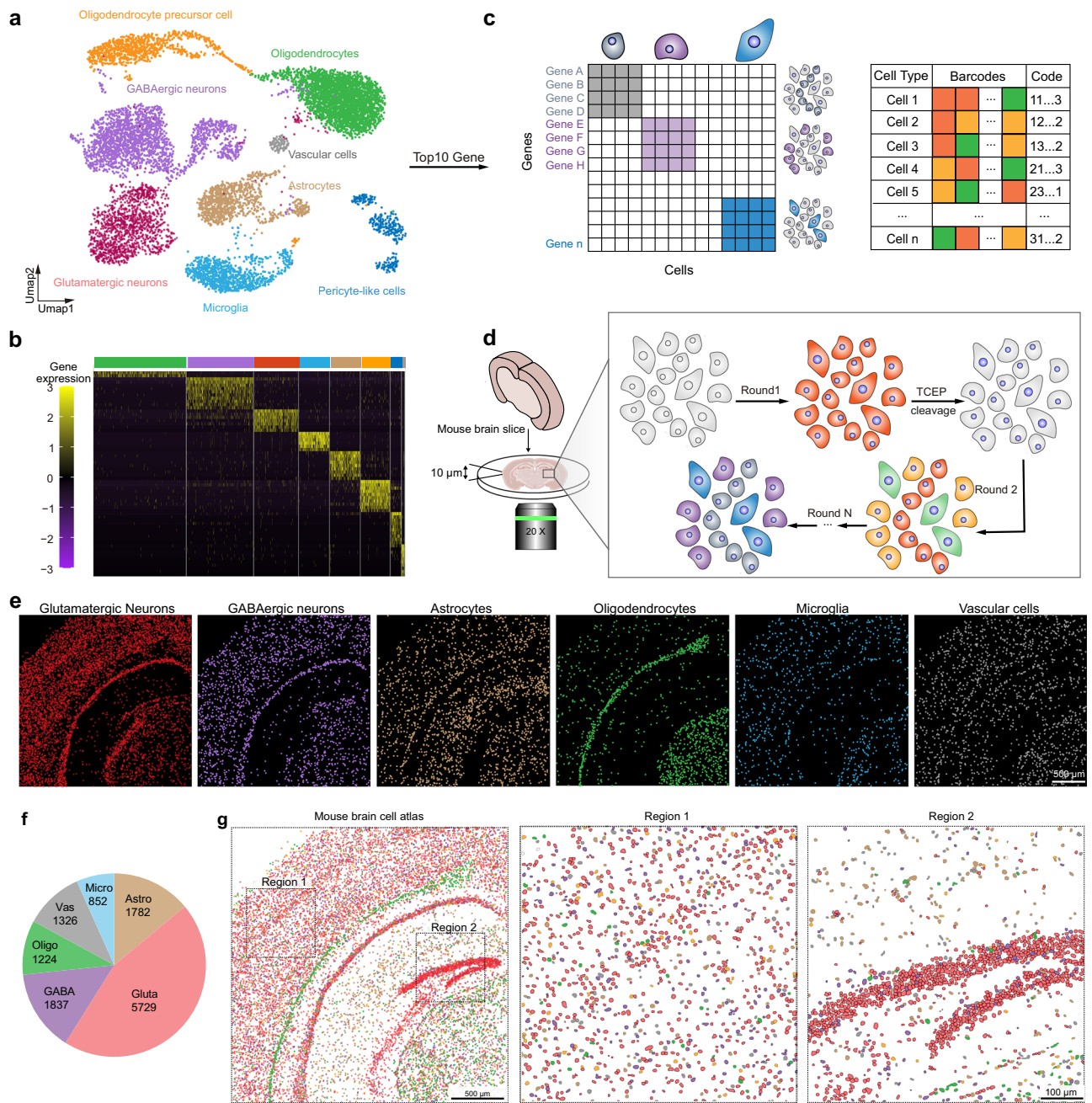

**Fig. 5 | TDDN-FISH mapping of cell types in the mouse brain. a** UMAP plot of different cell types in the mouse brain. **b** Heatmap of marker gene expression across annotated subclasses. **c** Cell-by-gene count matrix from single-cell RNA sequencing (scRNA-seq) can be used to cluster cell types, which are characterized by their unique gene expression profiles. **d** Diagram of spatial decoding of the mouse brain via two rounds of TDDN-FISH hybridization and imaging. **e** Spatial maps of the detected cells, colored by their cell types: Glutamatergic neurons (red), GABAergic neurons (purple), Astrocytes (brown), Oligodendrocytes (green), Microglia(cyan), Vascular cells (gray). Scale bar, 500 μm. **f** Cell type frequency detected by TDDN-FISH. **g** Spatial distribution of the six cell types overlaid, with a magnified view of a specific region. Scale bar, 500 μm.

rapid, sensitive, and enzyme-free platform for spatial RNA analysis. Capitalizing on the unique advantages of TDDN, TDDN-FISH offers ~8-fold faster single-round detection versus HCR-FISH and superior signal intensity over smFISH. This advancement enables high-resolution spatial transcriptomic profiling at both single-cell and subcellular levels, facilitating the exploration of intricate gene expression patterns in diverse biological contexts. We validated the effectiveness of TDDN-FISH by successfully mapping RNA expression patterns in both cultured cells and tissue samples, revealing complex gene expression dynamics with high precision. These capabilities highlight the

platform's potential to advance research in cellular heterogeneity, tissue organization, and disease pathology, offering a powerful tool for uncovering the spatial and molecular mechanisms underlying development, homeostasis, and disease.

In comparing TDDN-FISH with existing FISH methods (e.g., RCA-FISH, SABER, branched DNA FISH, RNAscope, ClampFISH, and MERFISH variants[10,39,42–45]), our analysis highlights its unique advantages in overcoming several critical limitations of current approaches. Unlike enzyme-dependent methods that often suffer from amplification variability, TDDN-FISH's enzyme-free operation

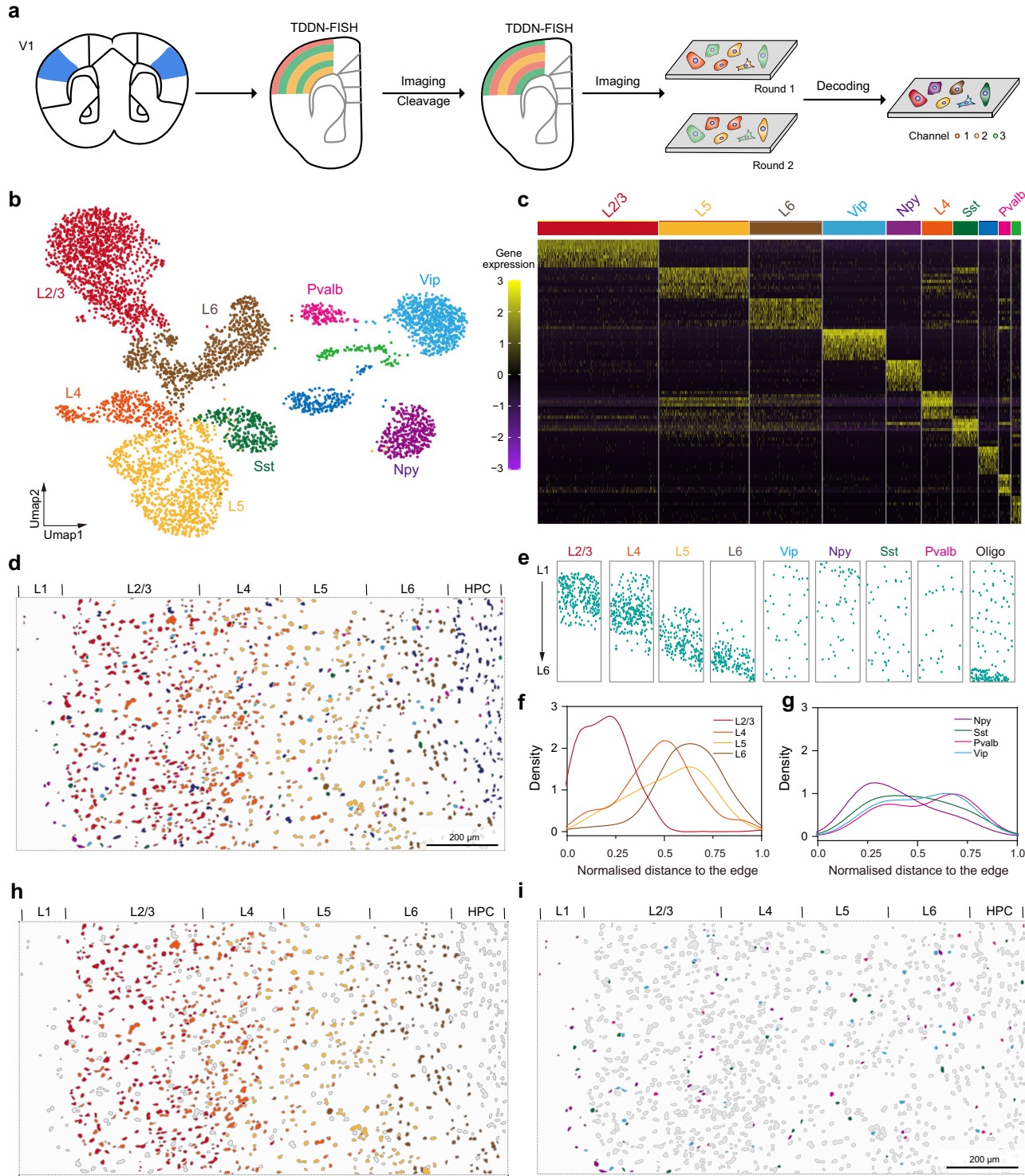

**Fig. 6 | Spatial registration of neuron subclasses in mouse primary visual cortex(V1) by TDDN-FISH in situ detection. a** Diagram of spatial decoding of neuronal subtypes in mouse V1 via two rounds of TDDN-FISH hybridization and imaging. **b** UMAP plot of all annotated excitatory neurons (L2/3, L4, L5, L6), inhibitory neurons (Pvalb, Vip, Npy, Sst). L1 to L6 denote the six layers of the neocortex; HPC refers to the hippocampus. Vip, Npy, Sst, and Pvalb represent four major types of inhibitory neurons. Oligo indicates oligodendrocytes. **c** Heatmap of marker gene expression across annotated subclasses. **d, e** TDDN-FISH accurately reproduced the spatial distribution of layer-specific cells. Scale bar, 200 µm. Kernel Density Estimate (KDE) of cell density for excitatory neurons (**f**) and inhibitory neurons (**g**) along the cortical depth. Spatial distribution of excitatory neurons (**h**) and inhibitory neurons(**i**). Scale bar, 200 µm. Source data are provided as a Source Data file.

ensures more consistent performance while achieving rapid detection within just 1 h. Its modular DNA nanostructure design not only enables parallel probe assembly for efficient hybridization but also facilitates color-coding and multi-round detection. Importantly,

TDDN-FISH demonstrates superior versatility by accommodating short RNA targets without requiring complex probe designs, while maintaining compatibility with standard fluorescence microscopy and offering potential for integration with expansion microscopy.

Collectively, these advantages position TDDN-FISH as a powerful solution that simultaneously addresses major challenges such as detection time, target length requirements, and cost-effectiveness in multiplexed RNA detection.

While TDDN-FISH surpasses many existing methods, challenges remain, particularly in imaging ultra-high-density RNA populations (e.g., high-abundance mRNA comparable to *ACTB*), where signal overlap can complicate analysis. Data from HeLa cell RNA-seq data show that although such highly expressed mRNA accounts for less than 0.1%, their dense distribution highlights the issue of signal overlap (Supplementary Fig. 9)[46]. Therefore, breaking through the resolution limit in ultra-high-density scenarios is a core optimization direction for advancing TDDN-FISH technology in complex biological applications. Future integration with super-resolution or expansion microscopy could enhance its resolving power for detecting densely packed RNA in complex tissues, unlocking new possibilities for studying intricate RNA interactions and spatial organization in heterogeneous biological systems. Although TDDN-FISH performs well in tissue sections up to 40 μm thick, diffusion limitations may affect probe penetration and hybridization efficiency in specimens thicker than 100 μm. Future work needs to explore enhancing tissue permeability using techniques like CLARITY and pressure-assisted hybridization to extend TDDN-FISH's applicability to deeper, more complex tissue architectures[47,48]. To boost accuracy and reliability, future work can incorporate Hamming-distance-encoded probe sets enabling single-error correction and double-error detection, and refine imaging algorithms and deploy advanced machine-learning tools for automated error identification and correction.

Additionally, we adopt a cell-type-centric approach using marker gene panels rather than individual gene detection for tissue sample imaging, inspired by FISHnCHIPs[37]. By simultaneously targeting co-expressed RNAs, we achieve enhanced signal intensity that enables robust cell typing even in challenging samples while permitting rapid whole-slide imaging at lower magnifications. The redundant marker design provides inherent tolerance to gene dropout across varying sample conditions. To address potential RNA overlap, we further utilized a normalization-based decoding algorithm that compares relative fluorescence intensities across channels within each region, assigning the strongest signal as the probable identity. This strategy combines the throughput advantages of lower-resolution imaging with the biological fidelity of multi-marker validation, overcoming key limitations of conventional single-molecule localization methods. Nonetheless, this approach reflects an inherent trade-off between imaging speed and quantitative resolution, as lower magnification improves throughput at the potential cost of spatial precision and per-molecule accuracy.

In conclusion, TDDN-FISH represents a paradigm shift in spatial transcriptomics, offering a versatile and high-performance platform for RNA detection across biological scales. Its unique combination of signal amplification, multiplexing, and speed addresses longstanding challenges in the field, paving the way for new discoveries in basic and translational research. As the technology continues to evolve, its integration with complementary imaging modalities and computational tools will further expand its applications, solidifying its role as a cornerstone of next-generation spatial omics. With its potential to transform our understanding of gene regulation and cellular interactions, TDDN-FISH is poised to advance biomedical research and clinical practice.

## Methods
### Animal samples
Six-week-old female Balb/c mice were obtained from Beijing Vital River Laboratory Animal Technology Co., Ltd. (Beijing, China). The mice were housed in specific-pathogen-free conditions at 25 °C and 40–60% relative humidity, with a 12 h light/dark cycle. Food and water were available ad libitum. All animal procedures were conducted in accordance with the Chinese Regulations for the Administration of Affairs Concerning Experimental Animals and were approved by the Committee for Animal Experimentation at Nankai University (SYXK [Jin] 2019-0003). As the primary aim of this study was to demonstrate the feasibility of the method in animal samples, sex was not considered in the experimental design or analysis.

### Tetrahedral DNA sequence design
The Tetrahedral DNA (Tn) monomer is assembled from Tna, Tnb, Tnc, and Tnd[22]. The sticky end of T0a is responsible for binding to the reading sequence of the primary probe, where T0a-hyb1-Cy5 indicates the first round of hybridization, binding to the Cy5-modified probe. The sticky ends of T0b, T0c, and T0d of tetrahedron T0 are identical and responsible for binding to the sticky ends of T1a of tetrahedron T1. The sticky ends of T1b, T1c, and T1d are identical and responsible for binding to the sticky end of T2a of tetrahedron T2. The sticky ends of T2a, T2b, and T2c are identical and are responsible for binding to dye-modified single-stranded nucleic acids (Cy5-ssDNA, Cy3-ssDNA, FAM-ssDNA). All sequences used were synthesized by Tsingke Biotechnology Co., Ltd. (Beijing, China). All sequences are listed in Supplementary Data 1.

### Synthesis of tetrahedral DNAs
To synthesize tetrahedral DNA monomers, four-component oligonucleotides of the Tn structure were mixed in equimolar concentration (1 μM each) in 1× TM buffer (50 mM Tris-HCl, 8 mM $MgSO_4$, pH 7.5). The procedures were performed as follows: denaturation at 95 °C for 5 min, and then cooled to 4 °C for 30 min. The final annealed products were stored at 4 °C. The final concentration of synthesized Tetrahedral DNA nanostructure was 1 μM[22].

### Agarose gel electrophoresis
Agarose gel electrophoresis was performed to investigate the synthesis of the products. All of the products, including ssDNA and Tetrahedral DNA monomers, were mixed with 6× DNA loading buffer. Electrophoresis was performed with 2% agarose gel in 1× Tris-acetate-EDTA (TAE, pH 8.3) at 110 V for 40 min. After electrophoresis, the samples were analyzed using a gel imaging system.

### Assembly of tetrahedral DNA dendritic nanostructures (TDDNs)
The prepared Tetrahedral DNA monomers were mixed step by step to prepare different generations of Tetrahedral DNA dendrimers. The sequences of Tetrahedral DNA monomers were specifically designed so that the hybridization of sticky ends could only happen between Tn and Tn+1. In each preparation step, three molar ratio Tn was mixed with one molar ratio Tn−1 to form Tn,n + 1. In a typical experiment, 5 μL T0 (the concentration is 1 μM for each strand as it was prepared in the paragraph above) was mixed with 15 μL T1 to prepare $TDDN_{0,1}$. The mixture was then incubated at room temperature for 1 h, allowing the hybridization of sticky ends to complete.

Meanwhile, T2 was mixed with fluorescent probes at a molar ratio of 1:3, allowing the dye-labeled DNA strands to bind to the sticky ends of T2. Subsequently, 45 μL of dye-labeled T2 was added to $TDDN_{0,1}$, and the mixture was incubated at room temperature for an additional 1 h to assemble $TDDN_{0,1,2}$[23].

### AFM imaging
AFM images were taken with a Bruker Dimension Icon microscope (Veeco Inc., USA). Freshly cleaved mica was treated with a 0.5% (v/v) APTES (Sigma-Aldrich) for 2 min, washed with Milli-Q water (18 MΩ cm), and dried with compressed air prior to nanostructure deposition. In all, the sample in 10 μL TM buffer was deposited on the mica surface and allowed to sit for 2 min for adsorption. Subsequently, the excess TM buffer was washed away at 1 min intervals to prevent salt

crystallization, repeating this process three to five times. The samples were imaged in tapping mode in solution using SNL-A tips.

## Primer probe design and synthesis

The primer probes used are 65 nucleotides (nt) long, comprising a 35-nt target sequence and a 30-nt readout sequence. The target sequences were designed using the OligoArray 2.0 program, which requires inputting various parameters to constrain the properties of the designed regions[49]. These parameters included a target region length of 35 nt, a melting temperature (Tm) for the properly hybridized probe greater than 70 °C, a minimum Tm of 72 °C for hybridization to potential off-target sequences, no internal secondary structures with a Tm lower than 76 °C, and no contiguous runs of the same nucleotide longer than six. These ranges were chosen to balance high stringency in probe binding with the ability to design sufficient distinct target regions to label each RNA. The readout sequences were designed based on MERFISH, utilizing 30-nt sequences[10]. All probe sequences are listed in Supplementary Data 1.

## Cell culture

Hela, A549, and MDCK cell lines were obtained from the Shanghai Cell Bank, Chinese Academy of Sciences. HeLa, A549, and MDCK cells were cultured in DMEM supplemented with 10% FBS and 1% penicillin–streptomycin at 37 °C under 5% $CO_2$. For single-molecule fluorescence in situ hybridization (smFISH) assays, cells were plated onto glass-bottom dishes and grown to 70–80% confluence prior to fixation.

## Transfection of cells

MDCK cells were transiently transfected with the plasmids encoding PCP-mCherry, and viral mRNA using jetPRIME transfection reagent (Polyplus) per manufacturer's protocol[25]. For 20 mm dishes, 1 μg of each plasmid was combined with 3 μL reagent and 100 μL buffer, incubated 10 min at room temperature, then added to cells. Complete medium was replaced at 6 h post-transfection.

## Virus infection

A549 cells were infected with influenza viruses using established protocols[50,51]. For synchronized entry, cells were chilled on ice for 5 min, then incubated with virus diluted in ice-cold infection medium (phosphate-buffered saline (PBS) supplemented with 1% bovine albumin (BSA) and 1% penicillin–streptomycin) for 60 min. After adsorption, cells were washed and immediately replenished with post-infection medium (DMEM supplemented with 0.3% BSA, 1% penicillin–streptomycin). Infection was initiated by transferring cultures to 37 °C.

## Mouse tissue sample preparation

The mice were euthanized, and their brains were rapidly collected and frozen in optimal cutting temperature compound (Tissue-Tek O.C.T. Compound) before being stored at −80 °C. The fresh frozen brain samples were subsequently sectioned into 10 μm-thick slices using a cryostat and mounted directly onto functionalized coverslips[37]. The sections were air-dried for 5 min at room temperature and then fixed with 4% paraformaldehyde (vol/vol) in 1× PBS for 15 min. After fixation, the samples were rinsed once with 1× PBS and either immediately permeabilized in 0.5% Triton X-100 in 1× PBS for 10 min at room temperature, permeabilized in 70% ethanol overnight at 4 °C, or stored at −80 °C. Since the objective was to demonstrate a technology, no sample-size estimate was performed.

## Coverslip functionalization

The 40 mm coverslips were first cleaned by immersing them in a potassium dichromate and concentrated sulfuric acid solution for 2 h. After thorough rinsing with water, they were washed with distilled water at least three times. Subsequently, the coverslips were soaked in 95% ethanol for 12 h and then allowed to air dry completely. A working solution of 50× Anti-Slice Escaping Agentia APES was freshly prepared by diluting it at a 1:50 ratio with acetone and used without delay. The dried coverslips were immersed in this solution for approximately 20–30 s, then removed and left briefly before being rinsed three times with distilled water to eliminate any residual unbound APES. Finally, the treated coverslips were stored in a clean, dust-free environment under dry conditions[31].

## TDDN-FISH procedure on cultured cells and tissue sample

The cells and tissue samples were washed twice with PBS and then fixed for 10 min with 4% formaldehyde in PBS at room temperature. The formaldehyde was aspirated off, and the samples were washed three times with PBS. They were then incubated with 5 μg/mL Proteinase K in diethyl pyrocarbonate (DEPC)-treated PBS buffer for 20 min. Next, the samples were transferred into permeabilization buffer (2× SSC, 0.5% Triton-X-100) at room temperature for 15 min, followed by three washes with 2× SSC buffer. The samples were then transferred to 70% ethanol at −20 °C overnight. Subsequently, they were incubated in hybridization buffer A (10% dextran sulfate, 2× SSC, 30% formamide, 20 mM RVC, 1 mg/mL yeast tRNA, 0.1% RNase-free BSA, and primary probes at 50 nM for each RNA) for 8–12 h at 37 °C or 42 °C. The samples were washed with hybridization wash buffer A (2× SSC, 30% formamide, 20 mM RVC) at 37 °C for 10 min (three times), followed by three washes with 2× SSC for 5 min at room temperature. The samples were then stored in 2× SSC at 4 °C. If prepared in an RNase-free environment, the samples could be stored in 2× SSC at 4 °C for several days. The next step involved hybridization of the tetrahedral DNA probes with the readout sequences of the primary probes. The samples were incubated in hybridization buffer B (10% dextran sulfate, 2× SSC, 5% formamide, 20 mM RVC, 1 mg/mL yeast tRNA, 0.1% RNase-free BSA, and TDDN-FISH probe) for 45 min to 1 h at room temperature. Finally, the samples were washed three times with hybridization wash buffer B (2× SSC, 5% formamide, 20 mM RVC) at room temperature, stained with DAPI (1 mg/mL) for 5 min, and either imaged immediately or stored in 2× SSC at 4 °C for several days[26]. All sequences are listed in Supplementary Data 1.

## smFISH and HCR-FISH procedure on cultured cells

The preparation of probes and the experimental procedures for smFISH and HCR-FISH were conducted following previously established protocols[6,52]. For smFISH, prepared samples were incubated in a hybridization solution containing 2× SSC, 30% (v/v) formamide, 2 mM VRC, 1 mg/mL yeast tRNA, 10% (w/v) dextran sulfate, and 10 nM target probes overnight (>8 h) at 37 °C. After hybridization, the samples were washed three times with a washing buffer consisting of 2× SSC, 30% (v/v) formamide, and 2 mM VRC. Subsequently, the samples were transferred to a hybridization buffer containing 2× SSC, 10% (v/v) formamide, 10% (w/v) dextran sulfate, 2 mM VRC, and 100 nM fluorescently labeled probes and incubated for 1 h at 37 °C. After this, the samples were washed three times with the same washing buffer and immediately imaged. For HCR-FISH, the prepared samples were first incubated in a hybridization solution composed of 5× SSC, 30% (v/v) formamide, 9 mM citric acid, 0.1% Tween 20, 50 μg/mL heparin, 10% (w/v) dextran sulfate, and 10 nM target probes, with the incubation lasting overnight (>8 h) at 37 °C. Following this, the samples were washed twice with a washing buffer containing 5× SSC, 30% (v/v) formamide, 9 mM citric acid, 0.1% Tween 20, and 50 μg/mL heparin for 5 min at room temperature. The samples were then subjected to an amplification process in a solution comprising 5× SSC, 0.1% Tween 20, 10% dextran sulfate, and 100 nM amplification probes, with this step lasting for over 8 h at room temperature. Finally, the samples were washed three times with 5× SSCT and immediately imaged. The detailed sequences

of the probes used for smFISH and HCR are provided in Supplementary Data 1.

## Multi-round TDDN-FISH imaging

As described above, the samples on the coverslip were fixed, permeabilized, and hybridized with primary probes[10]. After hybridization, they were stored in 2× SSC for further use. Then, the adhesive coverslip coated with the sample was assembled into a Bioptechs FCS2 flow chamber with temperature control. Fixed samples were first adhered onto adhesive coverslips (40 mm round, 0.15 mm thick); then a silicone gasket (40 mm round, 0.75 mm thick) with a central rectangle cavity was placed on the coverslip; and then the micro-aqueduct slide was placed on the gasket. The coverslip, gasket, and micro-aqueduct slide constitute a sandwich structure. Samples were in the cavity where buffers passed through. Fluidics was controlled via a peristaltic pump and set at a constant flow velocity of 500 μL/min (10 rpm). Multiple rounds of probe hybridization, imaging, and probe cleavage were performed as follows:

**First round hybridization.** To prevent bubble formation during hybridization, fill the FCS2 chamber with 2× SSC buffer. Subsequently, introduce Hybridization Buffer B (comprising 10% dextran sulfate, 2× SSC, 5% formamide, 20 mM RVC, 1 mg/mL yeast tRNA, 0.1% RNase-free BSA, and the TDDN-FISH probe) to completely fill the chamber. Perform hybridization at room temperature. After hybridization, wash away unbound probes using 2× SSC. All sequences are listed in Supplementary Data 1.

**Imaging.** Image acquisition was performed on a Nikon Ti2 inverted microscope configured with confocal laser scanning, a Yokogawa CSU-X1 spinning disk unit, and an Andor DU-897X EMCCD camera. The system employed three objective lenses: an SR HP APO TIRF 100× oil immersion objective (NA 1.49) for high-resolution cell imaging, an Apo TIRF 60× oil immersion objective (NA 1.49) for single-particle analysis in tissues, and a Plan Apo VC 20× air objective (NA 0.75) for large-area single-cell scans. Four-channel sequential excitation was implemented using 640 nm (diode laser), 561 nm (DPSS laser), 488 nm (diode laser), and 405 nm (diode laser) wavelengths, with corresponding emission signals isolated through bandpass filters (697/58 nm, 615/40 nm, 525/50 nm, and 447/60 nm, respectively). Continuous focal stability was maintained throughout all acquisitions via Nikon's Perfect Focus System.

**TCEP cleavage.** Cleavage was performed using 0.25 μM TCEP dissolved in 2× SSC, with the pH adjusted to 7.0 using NaOH, and incubated for 15 min to remove the signal from the previous round of hybridization. Then, the FCS2 chamber was washed with 2× SSC to ensure no residual TCEP affects the next round of hybridization probes.

## Imaging analysis

First, images were aligned to the position with the highest cross-correlation using custom MATLAB scripts to eliminate positional shifts occurring during multiple rounds of imaging. For cell samples, single-particle localization was performed using the TrackMate plugin in ImageJ to obtain the coordinates of fluorescence spots in the three channels. Each detected spot was assigned a color code based on the sequential appearance of different fluorescence signals across two imaging rounds. Specifically, Cy5 signals were defined as "1", Cy3 signals as "2", and FAM signals as "3". A two-digit code was then assigned to each individual spot to identify it as a single RNA molecule. For tissue samples, the Cellpose algorithm was used for cell segmentation to obtain cell contour masks[53]. Fluorescence intensity normalization was then applied to the images. Subsequently, each mask-defined region was analyzed independently using MATLAB scripts.

Given that cell-type-specific marker genes may exhibit low-level expression in other cell types, signals from all three channels (Cy5, Cy3, and FAM) may co-exist within a single mask region. Therefore, during the matching process, only the channel with the highest normalized fluorescence intensity among the three (Cy5, Cy3, FAM) was retained. If the normalized fluorescence intensity in the Cy5 channel was greater than that in the Cy3 and FAM channels, the mask region was assigned the value "1" in that imaging round. Similarly, if the Cy3 channel exhibited the highest intensity, the mask region was assigned "2", and if the FAM channel was dominant, it was assigned "3". By matching the results from two imaging rounds, each mask region was assigned a unique color code, ultimately generating a spatial cell atlas.

## Statistics and reproducibility

All statistical analyses were performed using GraphPad Prism 8.0 software. Group differences were assessed using unpaired two-tailed Student's $t$ test, one-way analysis of variance, and Tukey's post hoc multiple comparisons test. The thresholds for statistical significance were set as follows: ns (non-significant), $P > 0.05$, $*P < 0.05$; $**P < 0.01$; $***P < 0.001$.

TDDN-FISH has been successfully applied to various tissue types, demonstrating its reliability and reproducibility. The data presented are from a single experiment ($n = 1$). The experiments shown in Figs. 2B, 3D, G, 5, and 6 were each independently repeated at least once, with consistent results.

## Reporting summary

Further information on research design is available in the Nature Portfolio Reporting Summary linked to this article.

## Data availability

Source data are provided with this paper. The single-cell RNA-seq datasets of mouse brain utilized in this work can be accessed from the NCBI GEO repository under the accession number GSE115746. HeLa cell RNA-seq data can be accessed from the NCBI GEO repository under the accession number GSE123571. The data generated in this study are available from the corresponding author upon request and will be fulfilled within a week. Source data are provided with this paper.

## Code availability

The software to analyze TDDN-FISH data is available at Zenodo: https://doi.org/10.5281/zenodo.16993469.

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

## Acknowledgements

This research is financially supported by the National Natural Science Foundation of China (22293032) from D.-W.P., the National Natural Science Foundation of China (22374138) and Natural Science Foundation of Tianjin (24JCZDJC01240) from S.-L.L., and the Natural Science Foundation of Tianjin (23JCZDJC01240) from Z.-G.W.

## Author contributions

Y.-F.W. and S.-L.L. designed the research. Y.-F.W. performed most of the research. H.-J.C. and Z.-D.H. help to perform the experiments and analyze the data. Z.-G.W. and D.-W.P. provided their valuable guidance. Y.-F.W. and S.-L.L. wrote the paper. S.-L.L. conceptualized and supervised the study.

## Competing interests

The authors declare no competing interests.
