## [Transparent Peer Review file · Nature Communications]

Tetrahedral DNA Dendritic Nanostructure-Enhanced FISH (TDDN-FISH) for High-Speed, Sensitive Spatial Transcriptomics

Corresponding Author: Professor Shu-Lin Liu

Version 0:

Reviewer comments:

Reviewer #1

(Remarks to the Author)

I have carefully reviewed the paper titled "Tetrahedral DNA Dendritic Nanostructure-Enhanced FISH (TDDN-FISH) for High-Speed, Sensitive Spatial Transcriptomics" (No.NCOMMS-25-13026). The papers present a novel TDDN-FISH technology that uses tetrahedral DNA dendritic nanostructures for fluorescence in situ hybridization. This technology is an advancement as it allows for exponential signal amplification through programmable self-assembly. Compared to traditional methods like RCA-FISH and HCR-FISH, TDDN-FISH exhibits 2 - 4 times higher sensitivity and is 8 times faster in single-round detection. This innovation addresses the limitations of existing techniques, such as weak signals and slow detection, enabling more accurate detection of low-abundance RNAs. However, There are the following issues in the article that need to be carefully considered:

1. The authors did not provide the sequence information of RCA-FISH and HCR-FISH, nor the specific information of Toehold. It is uncertain whether the comparison was carried out under the same conditions, and this lack of clarity casts doubt on the comparison. According to previous literature reports, some classic FISH methods, such as RNAscope, can bind $20 \times 20 = 400$ - fold fluorescent probes at each probe locus. Pai - FISH can also provide an 8×16 - fold amplification signal, and RCA - FISH and HCR - FISH can at least provide the binding of more than 100 fluorescent molecules. However, in this article, each binding site only provides 27 binding sites for fluorescent molecules, indicating that the amplification signal is surely limited. It is unclear why the authors can achieve a stronger FISH result than RCA - FISH and HCR - FISH. Moreover, when the authors compared TDDN - FISH with traditional RCA - FISH and HCR - FISH, the experimental systems might not be completely consistent. Subtle differences in conditions such as the composition of the hybridization buffer, reaction temperature, and time can all affect the signal intensity. Although the paper mentions the optimization of key hybridization parameters for TDDN - FISH, it is difficult to ensure that all factors other than the core technology are exactly the same when compared with traditional methods. This may attribute the increase in the signal intensity of TDDN - FISH partly to the differences in the experimental system rather than the advantages of the technology itself.

2. After multiple assemblies of tetrahedral DNA, a relatively large - sized assembly will be formed, which is bound to affect the diffusion of molecules, especially for thicker tissues. Have the authors considered the limitations of this application? Can they provide an application example for thick tissues (such as tissues thicker than $40 \mu\text{m}$)?

3. The authors mentioned the comparison with MerFISH. MerFISH has limited signal amplification and can only target long mRNA molecules, such as mRNAs longer than 1500 bp. However, the authors did not provide the application comparison in this regard. What is the minimum size of the RNA that can be detected? Can it achieve the detection of miRNAs? Also, the authors used a 40 - nt - long binding sequence. How many such sequences are required for each mRNA molecule to achieve detection? If too many, will there be not many advantages in detecting the length of mRNAs compared with MerFISH? If too few, the signal and sensitivity will be affected.

Minor points:

1. The authors mentioned the use of disulfide - bond cleavage for multi - round imaging, but the relevant information is not

listed in the Table S1.

2. In addition, please provide all the sequence information related to hybridization for readers to repeat the experiments.

Reviewer #2

(Remarks to the Author)

Summary

This well written manuscript by Wang et al. describes a new way of amplifying smFISH signal. smFISH is a powerful method for spatial transcript quantification but suffers from low signal strength. This makes smFISH hard to apply in tissues with high autofluorescence and increases the imaging time substantially.

By using a branched DNA-origami structure in the form of multiple linked tetrahedrons, Wang et. al. introduce a way to substantially amplify the signal coming from a single RNA molecule. The branched tetrahedral probe called TDDN, consist of 3 layers of tetrahedrons, where each layer multiplies the number of potential fluorophores by 3. The first layer has an overhanging tail that can bind a complementary probe that has been hybridized to the target RNA. In this way the TDDN probes are re-usable for multiple different targets by exchanging the cheap RNA-binding probe. Another advantage of this amplification strategy is that unlike an RCA approach, the amplicon is still diffraction limited, which saves optical space. The authors developed a smFISH method called TDDN-FISH with these probes, and show that they can detect RNA in cultured cells and mouse brain tissue. Furthermore, they use these probes to perform multiple cycles of staining-imaging-label removal to break the colour-barrier and measure 9 genes in the same sample, which is used to image viral RNA developments during infection. Lastly, they use the probes to encode cell-types and measure their location in mouse brain sections.

Spatial transcriptomics is rapidly becoming the tool of choice to study cellular heterogeneity in health and disease, but there are still many limitations of these methods such as signal strength. Therefore, the amplification of the signal is an important topic and the introduction of TDDN probes is an exciting development.

However, in my opinion the current technical evaluation of TDDN-FISH is not sufficient to support the claims made in the manuscript. I am especially concerned about false positives and false negatives that this method could suffer from, and these should be thoroughly evaluated. To be clear, I am not saying that the method should be perfect, but any potential shortcomings must be extensively characterized. Therefore, I do not recommend publication of the manuscript in its current form but would like to invite the authors to thoroughly evaluate the performance of TDDN-FISH. If these concerns and the comments below are appropriately addressed I would recommend publication.

Major comments

How much does the TDDN probe amplify the signal compared to smFISH? I count $9 \times 3 = 27$ fluorophores per probe. This is less than the 48 probes used by Raj et al 2008. Is the signal stronger than smFISH?

In line 152 the authors write "Confocal imaging demonstrated a stepwise enhancement in fluorescence intensity corresponding to the sequential assembly of the TDDN layers". I do not understand why the intensity would increase stepwise. Maybe I misunderstood but I thought the fluorophore-probes could only bind to T2, so that there is no fluorescence to be expected in T0 and T0-T1. Could the authors please clarify this experiment and statement?

The experiments presented in figure 1d-f are not very convincing. This might be because the image resolution in the figure is not high enough, but I cannot see any signal spots that would correspond to individual RNA molecules. The TDDN-FISH signal in fig 1d seems to stain the entire cell (Similar in Fig 2e). Is TDDN-FISH not a single molecule technique? Is the ACTB expression too high so that optical crowding becomes a problem? If yes, could the authors please determine the upper limit of molecules per cell that can be detected with TDDN-FISH. Why are there no colonies or amplicons visible in the RCA-FISH and HCR-FISH images of Fig 1e? Please mention in the methods how the SNR is calculated. Quantifying the number of detected molecules for each of the methods would be a valuable performance metric. Please compare these with smFISH as high-sensitivity baseline.

My main concern with TDDN-FISH concerns false positives and false negatives, which are not sufficiently evaluated in my opinion. The strength of smFISH lies in the fact that there are multiple probes (>25) targeting the same RNA molecule. smFISH has low false positives, because a single probe that binds off-target will not generate sufficient signal strength to be counted. Furthermore, smFISH also has low false negatives because the multiple probes generate robustness.

TDDN-FISH uses only a single probe per target. This could affect the sensitivity because it is likely that not all mRNA molecules will be bound by a TDDN probe. Furthermore, it also increases the dependency on that single probe compared to smFISH. Because a specific part of the RNA molecule could be less accessible due to secondary structure, protein binding, PFA fixation etc. With only a single probe it is more likely that the RNA detection is affected by these effects. Could the authors discuss their experience with probe design? Are there cases where an RNA-binding probe did not work? Is it a requirement of TDDN-FISH that primary probes need to be validated before use?

From the gel results in figure S1a it is clear that not all DNA monomers will successfully assemble into a TDDN probe. As far as I can judge from the methods there is no purification of the fully assembled probes. Therefore, I think it would be highly likely that monomer T0a or partly assembled probe T0, both without any fluorophores, would bind to the primary probe, effectively blocking the detection and generating a false negative. Especially because these parts are smaller and will more easily diffuse into the tissue than the fully assembled probe. Could the authors comment on this and potentially try to purify

the fully assembled probes before performing the labelling.

Furthermore, the fact that a single TDDN is bright enough to generate a signal that could be counted as an RNA molecule, could give false positives in the case that probes binds aspecifically or gets stuck in the sample matrix. Could the authors quantify this by performing experiments in cells and tissue that omits the primary probe?

The experiment presented in Figure 1h, where TDDN is combined with PP7 detection of RNA is a nice experiment.

However, only one cell is shown and a quantification in terms of counted molecules that overlap or not overlap lacks. Please quantify the detected molecules in this experiment.

To further measure false negatives and positives, I strongly suggest the authors to perform an experiment where smFISH and TDDN-FISH are combined. Target a gene with a single TDDN probe and a smFISH probe set in two different colours. Then count the number of overlapping dots (True signal), number of smFISH only spots (False negatives), and number of TDDN only spots (False positives). Because I suspect more false positives in dense tissue slices, please perform this experiment both on cultured cells and in tissue sections and measure a substantial number of cells in both experiments. I would also suggest repeating this for multiple genes.

In line 200, the authors claim that the results of Figure 2a confirm the sensitivity of TDDN-FISH. However, these experiments do not prove the sensitivity, and actually, raise questions on the sensitivity in my opinion. I'm surprised that MALAT1 resolves into individual spots and that spots are also observed outside the nucleus. In most tissues that I have experience with, MALAT1 is the highest expressed gene with thousands of copies per cell predominantly located in the nucleus. Could the authors please comment on these surprising results? To quantify the sensitivity please compare the measurements with orthogonal techniques such as qPCR and/or single cell RNA-seq, to understand if TDDN-FISH measurements reflect accurate expression levels. Similarly, the results from Figure 2b could be compared to public MERFISH measurements in the mouse brain.

Regarding the multiplexing, could the authors comment on the efficiency of the barcoding strategy? From Figure 2g&h it seems that not all imaged spots are identified. Did this experiment include empty barcodes to measure the number of false positives in the barcoding? If not, could this experiment and quantification be added?

Why did the authors choose to encode cell types into their measurement, rather than measuring the 60 genes directly followed by clustering or mapping to the known cell types? It is a very interesting method to efficiently locate cell types and it is different from the common strategies. I think the manuscript could benefit from a more in-depth discussion on why this strategy was chosen compared to measuring gene expression first and identify cell type later. Does this method ever generate conflicting identities and how are they resolved? Also please cite FISHnCHIPs for this approach 038/s41467-024-46669-y

There are many other smFISH based amplification protocols apart from HCR and RCA. These include SABER, branched DNA FISH, RNA-Scope, ClampFISH, branched DNA MERFISH to name a few. These are not discussed in the manuscript. Could the authors include a discussion of these methods and how TDDN-FISH relates to them?

Minor comments

The claim that the multiplexing capability of TDDN-FISH performs Fluorophoresⁿ cycles on line 111, is a very strong claim and suggest that thousands of genes can be measured in a few cycles. However, due to optical crowding I doubt that this would work. I would suggest nuancing this point.

On line 115 the authors claim that TDDN-FISH is 2-4 times more sensitive than RCA or HCR, but I could not find the results supporting this claim. As the goal of these methods is to quantify transcripts, sensitivity should be the fraction of endogenous RNA molecules that the method can detect, and not the image signal to noise ratio.

On line 121 the authors claim that TDDN-FISH has unparalleled sensitivity, speed and resolution. However, the results presented in the manuscript do not suggest that these characteristics are substantially better than for instance seqFISH, MERFISH or ISS/Xenium. Please adjust these and similar superlative claims.

Line 169. Claims that TDDN-FISH is substantially faster than HCR or RCA. In this comparison the preparation of the TDDN probes, i.e. the amplification, is not taken into account, while this is the case for HCR and RCA. Therefore, I do not think this is a fair comparison. As example, RCA amplification can take overnight, but once amplified the detection can be done in one hour (<https://doi.org/10.1242/dev.202448>) similar to TDDN-FISH.

Figure 2b. Please provide a legend for the colours.

Figure 2c. Please indicate how many cells were measured.

Mouse gene names should have only their first letter capitalized and the rest lower case.

Line 209. I would not call *Satb2* and *Gad1* low expressed based on previous smFISH and scRNA-seq results. Please base low or high expression also on an orthogonal technique rather than TDDN-FISH alone.

Line 223, should also reference Lubeck et al. 2014 10.1038/nmeth.2892

Line 233. Should this refer to Figure 2f instead of Figure D? Please also add a legend to Figure 2f.

Line 249. Could the wording of “simultaneous dynamic imaging of the eight vRNA” be changed? Currently it could be confused with live imaging of 8 targets simultaneously.

Line 282. Please add the reference to the specific scRNA-seq dataset.

Figure 5e, why are there no oligodendrocytes observed inside the cortical layers? They should be there.

For Figure S1b it is not clear what the three different images are. Also is the color bar the Z-dimension? Please clarify this figure.

Supplemental methods

Line 33. Agarose gel electrophoresis. Why are the fully assembled probes not put on the gel?

Line 38. When and how are the fluorescent probes introduced to the TDDNs?

Line 115. Please indicate the buffer used for the Proteinase K, or was this done in pure water?

Line 148 Imaging. Please add details of the imaging system, including confocal system, camera or detector and light source. Please also include information on excitation power and exposure time.

Line 153. Was TCEP directly dissolved in water, or in 2x SSC?

Line 156. Will the analysis code be deposited somewhere like Github?

Figure S5b has poor image quality.

Please include the probe sequences used for all primary probes, RCA and HCR probes that are used in the manuscript.

Reviewer #3

(Remarks to the Author)

Current spatial transcriptomics technologies, such as RCA-FISH and HCR-FISH, have made some progress in RNA detection, but do still suffer from limitations such as complicated experimental processes, long time-consumption, and limited sensitivity. Wang et al. address these challenges by introducing the TDDN-FISH platform, which integrates tetrahedral DNA dendritic nanostructures (TDDNs) with fluorescence in situ hybridization (FISH). This innovative approach not only enhances the sensitivity and speed of RNA detection but also operates without enzymes, streamlining the experimental process and improving result reliability. This innovative approach not only enhances the sensitivity and speed of RNA detection but also operates without enzymes, streamlining the experimental process and improving result reliability. TDDN-FISH enables high-resolution RNA imaging at the single-molecule level with excellent multiplexing capability through efficient signal amplification mechanism. Color-coded barcoding technology further improves throughput and spatial resolution, making it possible to detect multiple RNAs simultaneously in complex biological samples. The authors demonstrate the broad applicability of TDDN-FISH through experiments in different biological systems such as cultured cells and tissue sections. In particular, TDDN-FISH demonstrated its strong potential for cellular localization of influenza virus RNA and RNA mapping of neuronal subtypes in the mouse brain.

Overall, Wang et al. present a promising tool for spatial transcriptomics. The TDDN-FISH platform offers significant advantages in sensitivity, speed, and multiplexing, while eliminating the need for enzymatic reactions, thereby simplifying experimental workflows. This technology opens new avenues for RNA research, particularly in high-resolution, single-molecule imaging and the analysis of complex biological samples.

Before publication, some details could benefit from further refinement. Nevertheless, the study's innovation, scientific rigor, and practical applicability make it a strong candidate for publication in Nature Communications.

Major points

1. In a multi-round hybridization experiment, DNA tetrahedra (T0) should have used tetrahedra with different sticky ends to bind to the primary sequence. How do the sticky ends ensure that they neither interfere with DNA tetrahedron assembly nor with specific binding to the primary sequence? The authors should describe clearly the design principles involved in this so that the method can be widely used.

2. In high-density RNA detection, signal overlap may occur, affecting single-molecule resolution. Would it be possible to integrate super-resolution microscopy techniques, such as STORM or SIM, or expansion microscopy to further enhance the spatial resolution of high-density RNA imaging? Additionally, in tissue-level imaging experiments, probe permeability may limit detection depth. Could the permeability of the TDDN structure be improved, or could tissue clearing techniques, such as CLARITY or iDISCO, be incorporated to enhance imaging performance in thick tissue samples?

3. In this study, each individual tetrahedral DNA dendritic is labeled with only one fluorescent dye, resulting in a single fluorescence signal during imaging. Would it be possible to customize the design so that a single tetrahedron carries two or

more dyes? This would not only generate individual signals for each dye but also produce colocalized signals from multiple dyes in each imaging cycle. Such an improvement could enhance the multiplexing capability of a single experiment, reduce the number of imaging rounds, and improve overall experimental efficiency.

4. The author mentioned that the hybridization lasts for approximately 45 minutes and the probe washing takes about 15 minutes. What is the rationale behind these time choices? Please provide quantitative data, such as when the fluorescence intensity reaches a plateau after 45 minutes of hybridization and 15 minutes of probe washing.

5. The author mentioned that the hybridization between the primary probe and endogenous mRNA requires more than 8 hours. Would it be possible to optimize the hybridization conditions to reduce this time to 6 or even 4 hours, thereby further enhancing the imaging speed of the TDDN-FISH system?

6. In the specificity validation experiments shown in Figure 1g-i, exogenously introduced viral RNA sequences were selected, which have low homology to the endogenous mRNA sequences of human cells, minimizing potential interference from sequence similarity. To further validate the specificity of the TDDN-FISH method, co-localization experiments using endogenous mRNA sequences should be conducted, including in complex tissue samples.

Minor points

7. In the supplementary materials, details of the RCA-FISH and HCR-FISH methods used in Fig. 1 should be provided, including the design methodology of RCA probes.

8. Some figure legends do not specify the number of cells analyzed in the statistical graphs. Please provide this information.

9. The colors in Figure 4g are not clear enough. Please provide an image with better distinguishability.

10. In Figure S5d, the images from the Allen Brain Atlas has a relatively low resolution. Please replace it with the high-resolution images.

Version 1:

Reviewer comments:

Reviewer #1

(Remarks to the Author)

The author has responded to most of the questions raised, but upon careful examination, the following issues in the author's response still need to be addressed:

1. When comparing similar technologies, the author chose to benchmark against smFISH, demonstrating that TDDN-FISH achieved less than a 1.5-fold increase in signal intensity. Regrettably, this improvement is not substantially significant. Additionally, the author appears to have only compared two methods that favor their technology, omitting comparisons with other high-signal methods such as RNA-scope or Pai-FISH. Methods like MerFISH or seqFISH, which are rooted in smFISH, necessitate gel preparation and tissue clearing to eliminate strong background interference in tissue samples due to the inherently weak smFISH signals. Given that this technology does not significantly enhance signal intensity compared to smFISH, how does it mitigate background interference in tissue samples, despite its ease of implementation in cultured cells? Furthermore, previous studies have shown that split primary probes (e.g., Split-FISH or HCR v3) significantly reduce background signals. In contrast, this study employs non-split linear probes, which may not only increase false-positive signals but also elevate background noise. We are curious about the strategies used by the author to minimize both background signals and false positives. Although the author addressed false positives, our experiments show that failing to remove proteins and lipids indeed allows probe binding to these components, leading to pronounced false positives.
2. The author utilizes two-bit encoding for multi-gene multiplexing without implementing error correction mechanisms analogous to those in MerFISH. How does the author distinguish incorrect gene codes, particularly in densely packed signal environments? Isn't signal decoding a significant challenge? Additionally, the author claims to have detected 60 genes (in Fig. 4) and 90 genes (in Fig. 5), yet probes in the the supplementary excel is listed only approximately 40 genes. Does this reflect failed detection or invalid signals for the remaining genes?
3. The signal spots in Fig. S3 appear disproportionately large, with some exceeding 1 μm in diameter—this is physiologically implausible. Is this due to saturated signals or excessive post-processing of contrast/brightness? In Figs. S2-c,d, Gad1 and Satb2 are characterized as low-expression genes, yet Gad1 is typically highly expressed. Does this indicate insufficient sensitivity? If these are low-expression signals, how were such a large number of statistical points obtained in panel d? Furthermore, the author used smFISH to validate TDDN-FISH rigor in Figs. S2-c,d. Given that smFISH inherently exhibits high false positives in tissues, did this step include protein/lipid removal protocols (e.g., as in MerFISH)? Is smFISH an appropriate validation method, and why not employ more rigorous approaches like Split-FISH or RNA-scope?
4. The manuscript contains numerous grammatical and formatting inconsistencies, including capitalization errors at sentence beginnings and improper use of "x" for multiplication, such as 2x SSC. The author is advised to conduct a thorough proofreading to rectify these issues.

Reviewer #2

(Remarks to the Author)

I would like to thank the authors for the revised manuscript that now contains much more technical detail and a more rigorous evaluation of TDDN-FISH. I'm happy to see the discussion and analysis of false negatives and false positives. I also want to apologize for my scepticism towards the MALAT1 results, I was unaware that MALAT1 could have this expression pattern, but the linked pi-FISH results were convincing. Most of my comments were addressed satisfactory for me, and I also generally agree with the responses to the questions of the other reviewers. I have a few last minor request and comments that I think are still missing from the manuscript. If the authors address these, I would recommend the manuscript for publication.

Could the authors clarify with which microscope objective each of the experiments was performed in the methods section? I was initially confused why the resolution seems to differ a bit between images. Could that be attributed to the difference in numerical aperture and magnification? If so, could the authors add a discussion that there is a trade-off between imaging speed and quantitative imaging?

In the revised manuscript the authors write: "challenges remain, particularly in imaging ultra-high-density RNA populations where signal overlap can complicate analysis". Could the authors analyse what portion of the genes would be excluded from being quantified by TDDN-FISH? I'm not expecting a very extensive analysis, but it would be interesting to at least check published single-cell RNA-seq data of Hela cells to see what portion of the genes has a similar and higher gene expression than ACTB to get a ballpark figure.

I was surprised to read that the authors were inspired by the FISHnCHIPS method but failed to cite that paper in their original manuscript, please ensure that all original work is appropriately cited.

The description of the microscope system should include the name of the laser and confocal system as well as the details of the filters used.

The description of the TCEP cleavage in the methods lacks the mention of SSC that was provided in the comments. Please also check that all other answers provided to the reviewers, were actually included in the manuscript.

I regret to read that the authors choose to not share their analysis code, and I would suggest the authors to reconsider. Making the code available will help others to build on your work and contributes to open and transparent science.

Figure S4d, the bottom row of the figure is duplicated.

In comment to the response to reviewer 3. I would disagree that STORM could work because STORM relies on single fluorophore blinking and is therefore incompatible with signal amplification.

Reviewer #3

(Remarks to the Author)

All the concerns have been carefully addressed. The manuscript can be considered for publication under its current status.

Version 2:

Reviewer comments:

Reviewer #1

(Remarks to the Author)

Sorry for the delayed reply. The author has responded to all the questions, and there are no other issues. I recommend this article for publication. Thanks!

Jie Yang

Reviewer #2

(Remarks to the Author)

I would like to thank the authors for revising the manuscript. With the implemented changes, I would now recommend the manuscript for publication.

Response to the reviewers' comments

Reviewer #1 (Remarks to the Author):

I have carefully reviewed the paper titled "Tetrahedral DNA Dendritic Nanostructure-Enhanced FISH (TDDN-FISH) for High-Speed, Sensitive Spatial Transcriptomics" (No.NCOMMS-25-13026). The papers present a novel TDDN-FISH technology that uses tetrahedral DNA dendritic nanostructures for fluorescence in situ hybridization. This technology is an advancement as it allows for exponential signal amplification through programmable self-assembly. Compared to traditional methods like RCA-FISH and HCR-FISH, TDDN-FISH exhibits 2 - 4 times higher sensitivity and is 8 times faster in single-round detection. This innovation addresses the limitations of existing techniques, such as weak signals and slow detection, enabling more accurate detection of low-abundance RNAs. However, There are the following issues in the article that need to be carefully considered:

<Response> We are truly grateful for your perceptive insights and affirmative feedback regarding the strengths of our research. Your recommendations have proven to be immensely beneficial, inspiring us to carry out pertinent experiments that have significantly elevated the caliber of our manuscript. Once again, thank you for your backing and precious suggestions.

1. The authors did not provide the sequence information of RCA-FISH and HCR-FISH, nor the specific information of Toehold. It is uncertain whether the comparison was carried out under the same conditions, and this lack of clarity casts doubt on the comparison. According to previous literature reports, some classic FISH methods, such as RNAscope, can bind $20 \times 20 = 400$ - fold fluorescent probes at each probe locus. Pai - FISH can also provide an 8×16 - fold amplification signal, and RCA - FISH and HCR - FISH can at least provide the binding of more than 100 fluorescent molecules. However, in this article, each binding site only provides 27 binding sites for fluorescent molecules, indicating that the amplification signal is surely limited. It is unclear why the authors can achieve a stronger FISH result than RCA - FISH and HCR - FISH. Moreover, when the authors compared TDDN - FISH with traditional RCA - FISH and HCR - FISH, the experimental systems might not be completely consistent. Subtle differences in conditions such as the composition of the hybridization buffer, reaction temperature, and time can all affect the signal intensity. Although the paper mentions the optimization of key hybridization parameters for TDDN - FISH, it is difficult to ensure that all factors other than the core technology are exactly the same when compared with traditional methods. This may attribute the increase in the signal intensity of TDDN - FISH partly to the differences in the experimental system rather than the advantages of the technology itself.

<Response> We sincerely appreciate the reviewer's meticulous attention to the details of our experimental design and comparison methods. The concerns raised about the sequence

information of RCA-FISH and HCR-FISH, as well as the specific information of Toehold, are indeed important for the clarity and reproducibility of our research. We apologize for the unclear description of the experimental method in the original manuscript.

- We acknowledge that a single TDDN-FISH probe binds only 27 fluorescent dye molecules, which may limit the signal amplification capability. In this work, to compensate for this, we employed three primary probes in our experimental setup, resulting in the binding of 81 fluorescent dye molecules per target site. This strategy was designed to enhance signal strength while mitigating the effects of RNA secondary structure formation and probe accessibility issues due to PFA fixation.
- Regarding the comparison with RCA-FISH, we recognize that factors such as enzyme activity, diffusion rates, and hybridization buffer composition can significantly influence signal intensities. Therefore, in the revised manuscript, we have removed the intensity comparison with RCA-FISH and instead added comparisons with smFISH. This data has been added into the revised manuscript as Figure 1c-f. For HCR-FISH, smFISH, and TDDN-FISH, we have ensured that the experimental conditions are identical, thereby making the results more comparable.
- We have included the detailed sequence information for smFISH and HCR-FISH in the supplementary materials (Supplementary Table S1 and S2). Additionally, we have provided the specific sequences and design principles of the Toehold probes in Supplementary Note S1. We believe that these additions will greatly enhance the transparency and replicability of our study.
- These revisions have been incorporated into the manuscript, and we thank you for your valuable suggestions.

2. After multiple assemblies of tetrahedral DNA, a relatively large - sized assembly will be formed, which is bound to affect the diffusion of molecules, especially for thicker tissues. Have the authors considered the limitations of this application? Can they provide an application example for thick tissues (such as tissues thicker than 40 μm)?

<Response> We appreciate the reviewer's kind suggestion. We fully agree that after multiple rounds of assembly, the size of the Tetrahedral DNA Dendritic Nanostructures (TDDNs) increases, which could potentially affect molecular diffusion, especially in thicker tissues.

- To address this potential limitation, we optimized the tissue permeabilization steps in all our imaging experiments, including extending the permeabilization time during sample preparation, to facilitate the penetration of larger nanostructures.
- As show in Figure R1-1, we successfully applied TDDN-FISH to mouse brain tissue sections with a thickness of 40 μm , and obtained uniform distribution and strong specific RNA signals throughout the tissue, demonstrating effective probe diffusion and hybridization even in relatively thick sections.

- This data has been added into the revised manuscript as Figure S3b. We have added clarifications regarding the tissue thickness in the revised manuscript to enhance the completeness and clarity of the work. We also acknowledge that for extremely thick tissues (e.g., >100 μm), diffusion barriers could become more prominent, and in future studies, we plan to explore strategies such as tissue clearing techniques (e.g., CLARITY) or pressure-assisted hybridization to further improve probe penetration. Once again, we sincerely thank the reviewer for this valuable suggestion, which has helped us further strengthen our study.

Figure R1-1. TDDN-FISH imaging of *Gad1* mRNA in 40 μm tissue sections.

3. The authors mentioned the comparison with MerFISH. MerFISH has limited signal amplification and can only target long mRNA molecules, such as mRNAs longer than 1500 bp. However, the authors did not provide the application comparison in this regard. What is the minimum size of the RNA that can be detected? Can it achieve the detection of miRNAs? Also, the authors used a 40 - nt - long binding sequence. How many such sequences are required for each mRNA molecule to achieve detection? If too many, will there be not many advantages in detecting the length of mRNAs compared with MerFISH? If too few, the signal and sensitivity will be affected.

<Response> We appreciate the reviewer' s kind suggestion. Our TDDN-FISH method requires a single 35-nt binding site to initiate signal amplification, which allows for the detection of short RNA molecules, including microRNAs (miRNAs).

- First, we acknowledge that in the data presented in our manuscript, we typically design and utilize 3 such sequences per mRNA target in cells, to avoid issues like poor probe accessibility and to increase the success rate.
- However, based on our experimental experience, more than 6 fluorophores are sufficient for single-molecule labeling (*Nano Today* 2021, 40, 101271). In our case, a TDDN probe

containing 27 fluorophores is fully capable of meeting the requirements for single RNA imaging in cells. To demonstrate this, we performed TDDN-FISH labeling and imaging of miRNA, specifically miR-21 (72 nucleotides in length), as shown in the Figure R1-2. This proves that our method can achieve imaging of very short RNAs.

- Thus, TDDN-FISH offers a distinct advantage in detecting short mRNAs, where MerFISH faces limitations. which generally requires dozens of probe-binding regions and is thus restricted to long transcripts. This data has been added into the revised manuscript as Figure S1h.

Figure R1-2. Detection of *miR-21* using TDDN-FISH in HeLa cells.

Minor points:

1. The authors mentioned the use of disulfide - bond cleavage for multi - round imaging, but the relevant information is not listed in the Table S1.

<Response> We are grateful to the reviewer's kind suggestions. We have corrected in the revised manuscript.

2. In addition, please provide all the sequence information related to hybridization for readers to repeat the experiments.

<Response> We are grateful to the reviewer's kind suggestions. All sequence information used in this work can be found in Supplementary Data.

Reviewer #2 (Remarks to the Author):

Summary

This well written manuscript by Wang et al. describes a new way of amplifying smFISH signal. smFISH is a powerful method for spatial transcript quantification but suffers from low signal strength. This makes smFISH hard to apply in tissues with high autofluorescence and increases the imaging time substantially.

By using a branched DNA-origami structure in the form of multiple linked tetrahedrons, Wang et al. introduce a way to substantially amplify the signal coming from a single RNA molecule. The branched tetrahedral probe called TDDN, consist of 3 layers of tetrahedrons, where each layer multiplies the number of potential fluorophores by 3. The first layer has an overhanging tail that can bind a complementary probe that has been hybridized to the target RNA. In this way the TDDN probes are re-usable for multiple different targets by exchanging the cheap RNA-binding probe. Another advantage of this amplification strategy is that unlike an RCA approach, the amplicon is still diffraction limited, which saves optical space.

The authors developed a smFISH method called TDDN-FISH with these probes, and show that they can detect RNA in cultured cells and mouse brain tissue. Furthermore, they use these probes to perform multiple cycles of staining-imaging-label removal to break the colour-barrier and measure 9 genes in the same sample, which is used to image viral RNA developments during infection. Lastly, they use the probes to encode cell-types and measure their location in mouse brain sections.

Spatial transcriptomics is rapidly becoming the tool of choice to study cellular heterogeneity in health and disease, but there are still many limitations of these methods such as signal strength. Therefore, the amplification of the signal is an important topic and the introduction of TDDN probes is an exciting development.

However, in my opinion the current technical evaluation of TDDN-FISH is not sufficient to support the claims made in the manuscript. I am especially concerned about false positives and false negatives that this method could suffer from, and these should be thoroughly evaluated. To be clear, I am not saying that the method should be perfect, but any potential shortcomings must be extensively characterized. Therefore, I do not recommend publication of the manuscript in its current form but would like to invite the authors to thoroughly evaluate the performance of TDDN-FISH. If these concerns and the comments below are appropriately addressed I would recommend publication.

<Response> Thank you very much for your recognition of the significance and innovation of our work. We also highly appreciate your attention to the reliability of the TDDN-FISH technique, which will be very conducive to the improvement of the quality of our manuscript.

- We fully agree that the assessment of false positives/negatives plays a crucial role in method validation. In response to your comments as follows, we have conducted the corresponding supplementary experiments. We hope that the revised manuscript will meet the requirements for publication. Thank you once again.

Major comments

How much does the TDDN probe amplify the signal compared to smFISH? I count $9 \times 3 = 27$ fluorophores per probe. This is less than the 48 probes used by Raj et al 2008. Is the signal stronger than smFISH?

<Response> We sincerely appreciate the reviewer's insightful question regarding the signal amplification mechanism, and we apologize for not clearly explaining these experimental details in our original manuscript.

- While a single TDDN-FISH probe delivers 27 fluorophores (9×3), which may seem fewer than the 48 fluorophores used in traditional smFISH (e.g., Raj et al., 2008). However, in practice, we typically employ three primary probes per mRNA target to ensure robust detection. This strategy minimizes false negatives caused by mRNA-protein interactions or secondary structures that may limit probe accessibility. Consequently, each mRNA molecule is labeled with 81 fluorophores—surpassing conventional smFISH in total dye density.
- Additionally, our experimental data on *ACTB* mRNA imaging confirmed that TDDN-FISH yields significantly stronger signal intensity than smFISH under identical conditions (Figure R2-1). This data has been added into the revised manuscript as Figure 1c-f.
- We regret any confusion caused by our initial incomplete description and have provided experimental details in the revised manuscript.

Figure R2-1. Comparison of hybridization efficiency between TDDN-FISH and smFISH by detecting *ACTB* mRNA in HeLa cells ($n = 30$). Scale bar, 100 μm .

In line 152 the authors write “Confocal imaging demonstrated a stepwise enhancement in fluorescence intensity corresponding to the sequential assembly of the TDDN layers”. I do not understand why the intensity would increase stepwise. Maybe I misunderstood but I thought the fluorophore-probes could only bind to T2, so that there is no fluorescence to be expected in T0 and T0-T1. Could the authors please clarify this experiment and statement?

<Response> Thank you for raising this important question, and we apologize for not clearly explaining these experimental details in our original manuscript.

- In this experiment, to demonstrate that fluorescence intensity increases with each additional layer of tetrahedral assembly, we specifically designed fluorescent DNA probes that could bind to the sticky ends of the TDDN₀ or TDDN_{0,1} structure. This approach enabled us to visualize and quantify the progressive signal enhancement at each assembly stage through confocal microscopy.
- We appreciate your attention to this detail and have provided additional clarification in revised manuscript.

The experiments presented in figure 1d-f are not very convincing. This might be because the image resolution in the figure is not high enough, but I cannot see any signal spots that would correspond to individual RNA molecules. The TDDN-FISH signal in fig 1d seems to stain the entire cell (Similar in Fig 2e). Is TDDN-FISH not a single molecule technique? Is the *ACTB* expression too high so that optical crowding becomes a problem? If yes, could the authors please determine the upper limit of molecules per cell that can be detected with TDDN-FISH. Why are there no colonies or amplicons visible in the RCA-FISH and HCR-FISH images of Fig 1e? Please mention in the methods how the SNR is calculated. Quantifying the number of detected molecules for each of the methods would be a valuable performance metric. Please compare these with smFISH as high-sensitivity baseline.

<Response> We sincerely appreciate the reviewer's thoughtful critique regarding the single-molecule resolution in our imaging experiments. We would like to address these important concerns as follows:

- TDDN-FISH is fundamentally a single-molecule technique, as clearly demonstrated in Figure 2a-d where individual mRNA molecules are resolved for medium-to-low abundance targets. The apparent whole-cell staining observed for *ACTB* mRNA (Fig. 1d-f) results from the exceptionally high expression level of this housekeeping gene, leading to signal overlap that prevents resolution of individual molecules.
- For optical crowding limitations, we acknowledge this represents a technical limitation common to all single-molecule FISH methods. As referenced (*Nat. Commun.* 2023; 14:443), similar challenges occur with pi-FISH for highly expressed genes.
- Additionally, for RCA/HCR-FISH observations, the lack of visible colonies/amplicons in Figure 1e similarly reflects *ACTB*'s high abundance.
- We agree that SNR comparison provides limited value for evaluating performance. Instead, quantitative measurement of detected molecule counts serves as a more meaningful and valuable metric. Accordingly, in the revised manuscript, we have added direct comparisons between TDDN-FISH and smFISH, using molecular counts as the high-sensitivity baseline for evaluation.
- Many thanks for the insightful suggestions.

My main concern with TDDN-FISH concerns false positives and false negatives, which are not sufficiently evaluated in my opinion. The strength of smFISH lies in the fact that there are multiple probes (>25) targeting the same RNA molecule. smFISH has low false positives, because a single probe that binds off-target will not generate sufficient signal strength to be counted. Furthermore, smFISH also has low false negatives because the multiple probes generate robustness.

TDDN-FISH uses only a single probe per target. This could affect the sensitivity because it is likely that not all mRNA molecules will be bound by a TDDN probe. Furthermore, it also increases the dependency on that single probe compared to smFISH. Because a specific part of the RNA molecule could be less accessible due to secondary structure, protein binding, PFA fixation etc. With only a single probe it is more likely that the RNA detection is affected by these effects. Could to authors discuss their experience with probe design? Are there cases where an RNA-binding probe did not work? Is it a requirement of TDDN-FISH that primary probes need to be validated before use?

<Response> We greatly appreciate the reviewer's insightful concerns regarding false positives/negatives in TDDN-FISH, which are crucial considerations for any single-molecule detection method. We would like to address these points systematically:

- While TDDN-FISH can theoretically work with a single probe, our standard practice is to use 3-5 primary probes per target (similar to RNAscope and HCR-FISH), each targeting distinct transcript regions. This approach can reduce false negatives by overcoming accessibility issues (secondary structures, protein binding, fixation artifacts), and maintain detection robustness comparable to smFISH.
- The sticky end sequences of TDDN probes were designed according to the literature (*Science*, 2015; 348, aaa6090), including local BLAST alignment to minimize potential cross-hybridization with endogenous mRNAs or off-target sequences. Through iterative computational screening and experimental validation (via agarose gel electrophoresis), we confirmed that the selected sequences: (1) preserved tetrahedral nanostructure stability during assembly, (2) exhibited negligible non-specific binding to cellular RNAs, and (3) maintained high binding specificity to their intended targets in fluorescence assays. This comprehensive design-validation approach guarantees both the structural fidelity of the DNA nanostructures and the biological specificity required for reliable TDDN-FISH applications.
- We sincerely apologize for the lack of clarity regarding our methodology in the original manuscript and have now incorporated detailed revisions to address this issue.

From the gel results in figure S1a it is clear that not all DNA monomers will successfully assemble into a TDDN probe. As far as I can judge from the methods there is no purification of the fully assembled probes. Therefore, I think it would be highly likely that monomer T0a or partly assembled probe T0, both without any fluorophores, would bind to the primary probe, effectively blocking the detection and generating a false negative. Especially because these parts are smaller and will more easily diffuse into the tissue than the fully assembled probe. Could the authors comment on this and potentially try to purify the fully assembled probes before performing the labelling.

Furthermore, the fact that a single TDDN is bright enough to generate a signal that could be counted as an RNA molecule, could give false positives in the case that probes binds aspecifically or gets stuck in the sample matrix. Could the authors quantify this by performing experiments in cells and tissue that omits the primary probe?

<Response> We thank the reviewer for raising these important concerns.

- We thank the reviewer for raising these important concerns.
- We acknowledge the reviewer's valid point regarding the importance of DNA tetrahedron purification to minimize false negatives and fully recognize that additional purification (e.g., via SEC or PAGE) could further enhance performance.
- However, as shown in Figure S1a (also presented in Figure R2-2), gel electrophoresis analysis of the unpurified assembly products clearly demonstrated the successful formation of DNA tetrahedral nanostructures (TDDNs), and the nearly disappearance of the T0a monomer band post-assembly. Our current TDDN probe assembly protocol strictly maintains a 1:1:1:1 ratio of the four DNA strands. Moreover, TDDN-FISH has demonstrated an accuracy of over 90% in quantitative analysis, using smFISH as a benchmark. These results indicated that the current unpurified TDDN probes are sufficient for meeting the basic

requirements of existing FISH imaging applications. We have explicitly added this discussion point in the revised manuscript.

- Additionally, to accurately assess the potential for false positives arising from nonspecific binding, we meticulously designed and conducted control experiments. In these experiments, we omitted the primary probe in both cultured cells (Figure S1c) and frozen tissue samples (Figure R2-3). The results revealed that the resulting fluorescence signals were virtually undetectable. This clearly demonstrated that under our current hybridization and washing conditions, the nonspecific binding of TDDN probes is negligible. This finding underscores the high specificity of our experimental approach and provides a robust foundation for our subsequent research. This data has been added into the revised manuscript as Figure S1c.

Figure R2-2. Successful assembly of Tetrahedral DNA monomer (T0, T1, T2) demonstrated by gel electrophoretic analysis. Each DNA tetrahedron consists of 4 single strands. For example, T0 is composed by T0a, T0b, T0c and T0d, and the assemble process examined by gel electrophoretic analysis.

Figure R2-3. The specificity of TDDN-FISH validated with multiple negative controls, including no treatment, no Primary probes and no TDDN probes. Scale bars, 100 μ m.

The experiment presented in Figure 1h, where TDDN is combined with PP7 detection of RNA is a nice experiment. However, only one cell is shown and a quantification in terms of counted molecules that overlap or not overlap lacks. Please quantify the detected molecules in this experiment.

To further measure false negatives and positives, I strongly suggest the authors to perform an experiment where smFISH and TDDN-FISH are combined. Target a gene with a single TDDN probe and a smFISH probe set in two different colours. Then count the number of overlapping dots (True signal), number of smFISH only spots (False negatives), and number of TDDN only spots (False positives). Because I suspect more false positives in dense tissue slices, please

perform this experiment both on cultured cells and in tissue sections and measure a substantial number of cells in both experiments. I would also suggest repeating this for multiple genes.

<Response> We sincerely appreciate your valuable suggestions.

- We have replaced the single-cell image with a multi-cell image in Figure 1h. We also systematically evaluated method performance by assessing false-negative rates (detected through PP7-mCherry signal only), false-positive rates (identified by TDDN-FISH signal only), and overall co-localization efficiency (Figure R2-4a-c). This data has been added into the revised manuscript as Figure 1h and i.
- Meanwhile, we tested two genes, *POLR2A* and *HSP70*, in cell samples and *Satb2* and *Gad1* in tissue samples using TDDN FISH, and compared them with smFISH to obtain false-positive and false-negative data for different genes and different samples. The results demonstrated that the TDDN technique exhibited low false-positive and false-negative rates in both tissue and cell samples, achieving an accuracy of over 90% in both cases (Figure R2-4d-f).
- This data has been added into the revised manuscript as Figure S2c-e.

Figure R2-4. Colocalization analysis of the RNA signals from TDDN-FISH and smFISH. a. Dual-channel confocal images of target RNA labelled by TDDN-FISH and PCP-mCherry. Scale bar, 10 μ m. b. 2D frequency scatterplots of the signals from TDDN-FISH and PCP-mCherry shown in (a). c. Quantitative analysis of the colocalization signals (yellow), false-positive signals (green) and false-negative signals (red) shown in (a) (n = 5). d. Dual-channel confocal images of target RNA signals labelled by TDDN-FISH and smFISH. e. 2D frequency scatterplots of the signals from TDDN-FISH and smFISH shown in (d). f. Quantitative analysis of the colocalization signals (yellow), false-positive signals (green) and false-negative signals (red) shown in (d) (n = 5).

In line 200, the authors claim that the results of Figure 2a confirm the sensitivity of TDDN-FISH. However, these experiments do not prove the sensitivity, and actually, raise questions on the sensitivity in my opinion. I'm surprised that MALAT1 resolves into individual spots and that spots are also observed outside the nucleus. In most tissues that I have experience with, MALAT1 is the highest expressed gene with thousands of copies per cell predominantly located in the nucleus. Could the authors please comment on these surprising results? To quantify the sensitivity please compare the measurements with orthogonal techniques such as qPCR and/or single cell RNA-seq, to understand if TDDN-FISH measurements reflect accurate expression levels.

<Response> We appreciate the reviewer's insightful comments regarding the distribution of MALAT1.

- MALAT1 is commonly recognized as a highly expressed gene, primarily located in the nucleus. However, previous studies using smFISH and qPCR have confirmed that *MALAT1* can be present in both the nucleus and cytoplasm in HeLa cells (*Genome Biol.*, 2015; 16, 20; *Nat. Commun.*, 2023; 14, 443).
- Our experiments confirmed that *MALAT1* lncRNA is distributed in both the nucleus and the cytoplasm, with a higher abundance in the nucleus compared to the cytoplasm, consistent with previously reported findings in the literature (Figure R2-5).

Figure R2-5 TDDN-FISH used to detect *MALAT1* lncRNA in HeLa cells. The white lines in the image represent the nuclear contours. Scale bar, 10 μ m.

Why did the authors choose to encode cell types into their measurement, rather than measuring the 60 genes directly followed by clustering or mapping to the known cell types? It is a very interesting method to efficiently locate cell types and it is different from the common strategies. I think the manuscript could benefit from a more in-depth discussion on why this strategy was chosen compared to measuring gene expression first and identify cell type later. Does this method ever generate conflicting identities and how are they resolved? Also please cite FISHnCHIPs for this approach 038/s41467-024-46669-y

<Response> We thank the reviewer for the thoughtful comment and for recognizing the novelty of our cell-type encoding strategy.

- In recent years, spatial transcriptomics technologies such as MERFISH, HybISS, and STARmap have greatly advanced the resolution of gene expression mapping. However, these methods typically rely on single-molecule localization and require high-magnification microscopy to detect and decode signals, which can make large-scale tissue imaging time-consuming and technically demanding. Inspired by the concept introduced in FISHnCHIPs (DOI: [10.1038/s41467-024-46669-y]), we adopted a cell-type-centric strategy by encoding cell types through marker gene panels rather than detecting each of the 60 genes individually. This approach offers several practical and technical advantages:
 1. By targeting multiple co-expressed marker genes simultaneously for a specific cell type, we greatly increase the fluorescence signal per cell. This improves contrast, reduces background, and enhances the robustness of cell type identification—especially in tissues with poor RNA preservation or low-abundance transcripts.
 2. The elevated signal intensity allows us to use lower magnification microscopy for whole-slide scanning, significantly boosting imaging throughput and enabling rapid acquisition of large tissue areas—a major limitation of traditional single-molecule localization approaches.
 3. Since some RNA targets may be affected by RNA degradation, secondary structure, or crosslinking (e.g., due to PFA fixation), relying on a redundant marker set increases tolerance to gene dropout and ensures more stable classification across various sample conditions, including clinical specimens.
- Regarding the potential for conflicting identities, we acknowledge that marker genes are not always strictly exclusive to one cell type. To mitigate misclassification, we implemented a normalization-based decoding algorithm that compares relative fluorescence intensity across imaging channels within each mask region. After normalization, the color channel with the strongest signal is assigned as the most probable identity for that region, allowing us to robustly resolve overlapping marker expression patterns.
- The related discussion and the reference have been added in the revised manuscript.

There are many other smFISH based amplification protocols apart from HCR and RCA. These include SABER, branched DNA FISH, RNA-Scope, ClampFISH, branched DNA MERFISH to name a few. These are not discussed in the manuscript. Could the authors include a discussion of these methods and how TDDN-FISH relates to them?

<Response> We are grateful to the reviewer's kind suggestions.

- In comparing TDDN-FISH with existing FISH methods (e.g., RCA-FISH, SABER, branched DNA FISH, RNAscope, ClampFISH, and MERFISH variants), our analysis highlights its unique advantages in overcoming several critical limitations of current approaches. Unlike enzyme-dependent methods that often suffer from amplification variability, TDDN-FISH's enzyme-free operation ensures more consistent performance while achieving rapid detection within just one hour. Its modular DNA nanostructure design not only enables parallel probe assembly for efficient hybridization but also facilitates color-coding and multi-round detection. Importantly, TDDN-FISH demonstrates superior versatility by accommodating short RNA targets without requiring complex probe designs, while maintaining compatibility

with standard fluorescence microscopy and offering potential for integration with expansion microscopy. Collectively, these advantages position TDDN-FISH as a powerful solution that simultaneously addresses major challenges such as detection time, target length requirements, and cost-effectiveness in multiplexed RNA detection.

- The discussion has been added into the revised manuscript.

Minor comments

The claim that the multiplexing capability of TDDN-FISH performs Fluorophoresⁿ cycles on line 111, is a very strong claim and suggest that thousands of genes can be measured in a few cycles. However, due to optical crowding I doubt that this would work. I would suggest nuancing this point.

<Response> We agree with your opinion. We have removed the sentence in the revised manuscript and addressed this point in the Discussion section.

On line 115 the authors claim that TDDN-FISH is 2-4 times more sensitive than RCA or HCR, but I could not find the results supporting this claim. As the goal of these methods is to quantify transcripts, sensitivity should be the fraction of endogenous RNA molecules that the method can detect, and not the image signal to noise ratio.

<Response> We sincerely appreciate the reviewer's valuable feedback. We have revised the inappropriate description. Here, we primarily intend to highlight the variations in imaging intensity.

On line 121 the authors claim that TDDN-FISH has unparalleled sensitivity, speed and resolution. However, the results presented in the manuscript do not suggest that these characteristics are substantially better than for instance seqFISH, MERFISH or ISS/Xenium. Please adjust these and similar superlative claims.

<Response> We are grateful to the reviewer's kind suggestions. We have corrected in the revised manuscript.

Line 169. Claims that TDDN-FISH is substantially faster than HCR or RCA. In this comparison the preparation of the TDDN probes, i.e. the amplification, is not taken into account, while this is the case for HCR and RCA. Therefore, I do not think this is a fair comparison. As example, RCA amplification can take overnight, but once amplified the detection can be done in one hour (<https://doi.org/10.1242/dev.202448>) similar to TDDN-FISH.

<Response> We appreciate the reviewer's valuable suggestions. It should be clarified that the assembly process of TDDN is carried out in parallel with the hybridization time of the primary probe and intracellular mRNA. Therefore, we don't need extra time for signal amplification. We can directly add the TDDN probe for detection.

Figure 2b. Please provide a legend for the colours.

<Response> We are grateful to the reviewer's kind suggestions. We have added the information in the revised manuscript.

Figure 2c. Please indicate how many cells were measured.
<Response> We are grateful to the reviewer's kind suggestions. We have added the information in the revised manuscript.

Mouse gene names should have only their first letter capitalized and the rest lower case.
<Response> We are grateful to the reviewer's kind suggestions. We have corrected in the revised manuscript.

Line 209. I would not call *Satb2* and *Gad1* low expressed based on previous smFISH and scRNA-seq results. Please base low or high expression also on an orthogonal technique rather than TDDN-FISH alone.

<Response> We fully agree with this insightful observation and have modified the text in the revised manuscript to clarify that the expression levels of *Satb2* and *Gad1* are relatively lower compared with *ACTB*.

Line 223, should also reference Lubeck et al. 2014 10.1038/nmeth.2892
<Response> We are grateful to the reviewer's kind suggestions. We have added the reference in the revised manuscript.

Line 233. Should this refer to Figure 2f instead of Figure D? Please also add a legend to Figure 2f.
<Response> We are grateful to the reviewer's kind suggestions. We have corrected in the revised manuscript.

Line 249. Could the wording of "simultaneous dynamic imaging of the eight vRNA" be changed? Currently it could be confused with live imaging of 8 targets simultaneously.
<Response> We are grateful to the reviewer's kind suggestions. We have corrected the description as "simultaneous imaging of the eight vRNA" in the revised manuscript.

Line 282. Please add the reference to the specific scRNA-seq dataset.
<Response> We are grateful to the reviewer's kind suggestions. We have added the references in the revised manuscript.

Figure 5e, why are there no oligodendrocytes observed inside the cortical layers? They should be there.

<Response> We sincerely thank the reviewer for raising this important point.

- Upon re-examining our data with more comprehensive cortical sampling, we have now identified oligodendrocytes across all cortical layers (I-VI), as shown in the updated Figure 5e. The initial apparent absence resulted from analyzing limited regions that happened to have lower oligodendrocyte density.

For Figure S1b it is not clear what the three different images are. Also is the color bar the Z-dimension? Please clarify this figure.

<Response> We are grateful to the reviewer's kind suggestions.

- We have added the detail information in the revised manuscript. Figure S1b shows representative AFM images of the nanostructures formed at different stages. From left to right, the images correspond to the nanostructures TDDN₀, TDDN_{0,1}, and TDDN_{0,1,2}, respectively. The color bar represents the height along the Z-dimension.

Supplemental methods

Line 33. Agarose gel electrophoresis. Why are the fully assembled probes not put on the gel?
<Response> Conventional agarose gel can effectively separate linear DNA. However, due to not conforming to the charge/mass ratio migration law of linear DNA, non-linear DNA nanostructures are prone to band smearing and difficulty in distinguishing multimers, so we did not put them on the gel. Currently, there are many reports on the gel electrophoresis characterization of individual DNA tetrahedrons. However, more complex structures are mostly characterized by AFM.

Line 38. When and how are the fluorescent probes introduced to the TDDNs?
<Response> Thank you for your question.

- In our TDDN assembly protocol, fluorescent labeling is accomplished through a two-step hierarchical assembly process: (1) First, dye-modified single-stranded DNA probes are hybridized to T2 tetrahedral DNA nanostructures; (2) These pre-labeled T2 units are then assembled with T0 and T1 components to form the complete TDDN probe.
- We have added this information in the revised supporting information.

Line 115. Please indicate the buffer used for the Proteinase K, or was this done in pure water?
<Response> Thank you for your question. Proteinase K was dissolved in diethyl pyrocarbonate (DEPC)-treated PBS buffer.

Line 148 Imaging. Please add details of the imaging system, including confocal system, camera or detector and light source. Please also include information on excitation power and exposure time.
<Response> We are grateful to the reviewer's kind suggestions. We have added the information in the revised manuscript.

Line 153. Was TCEP directly dissolved in water, or in 2x SSC?

<Response> Thank you for your question. TCEP was dissolved in 2× SSC and the pH was adjusted to 7.0 using NaOH to ensure optimal cleavage efficiency of disulfide bonds.

Line 156. Will the analysis code be deposited somewhere like Github?

<Response> Since the code implementation was not a primary innovation of this work and was mainly adapted from existing spatial transcriptomics methods, we have chosen not to host it on GitHub. However, researchers interested in obtaining the code are welcome to contact us directly, and we would be happy to share the relevant materials to facilitate their work.

Figure S5b has poor image quality.

<Response> We are grateful to the reviewer's kind suggestions. We have changed the images in the revised manuscript.

Please include the probe sequences used for all primary probes, RCA and HCR probes that are used in the manuscript.

<Response> We are grateful to the reviewer's kind suggestions. We have added the sequence information in the Supplementary Data of the revised manuscript.

Reviewer #3 (Remarks to the Author):

Current spatial transcriptomics technologies, such as RCA-FISH and HCR-FISH, have made some progress in RNA detection, but do still suffer from limitations such as complicated experimental processes, long time-consumption, and limited sensitivity. Wang et al. address these challenges by introducing the TDDN-FISH platform, which integrates tetrahedral DNA dendritic nanostructures (TDDNs) with fluorescence in situ hybridization (FISH). This innovative approach not only enhances the sensitivity and speed of RNA detection but also operates without enzymes, streamlining the experimental process and improving result reliability. This innovative approach not only enhances the sensitivity and speed of RNA detection but also operates without enzymes, streamlining the experimental process and improving result reliability. TDDN-FISH enables high-resolution RNA imaging at the single-molecule level with excellent multiplexing capability through efficient signal amplification mechanism. Color-coded barcoding technology further improves throughput and spatial resolution, making it possible to detect multiple RNAs simultaneously in complex biological samples. The authors demonstrate the broad applicability of TDDN-FISH through experiments in different biological systems such as cultured cells and tissue sections. In particular, TDDN-FISH demonstrated its strong potential for cellular localization of influenza virus RNA and RNA mapping of neuronal subtypes in the mouse brain.

Overall, Wang et al. present a promising tool for spatial transcriptomics. The TDDN-FISH platform offers significant advantages in sensitivity, speed, and multiplexing, while eliminating the need for enzymatic reactions, thereby simplifying experimental workflows. This technology opens new avenues for RNA research, particularly in high-resolution, single-molecule imaging and the analysis of complex biological samples.

Before publication, some details could benefit from further refinement. Nevertheless, the study's innovation, scientific rigor, and practical applicability make it a strong candidate for publication in Nature Communications.

<Response> We sincerely appreciate your insightful understanding and positive feedback on the merits of our research. Your suggestions have been highly valuable, prompting us to conduct relevant experiments that have greatly enhanced the quality of our manuscript. We hope that the revised manuscript now fully meets the publication requirements. Thank you again for your support and valuable input.

Major points

1. In a multi-round hybridization experiment, DNA tetrahedra (T0) should have used tetrahedra with different sticky ends to bind to the primary sequence. How do the sticky ends ensure that they neither interfere with DNA tetrahedron assembly nor with specific binding to the primary sequence? The authors should describe clearly the design principles involved in this so that the method can be widely used.

<Response> Thank you for your valuable comments.

- Our DNA tetrahedron (T0) design incorporated carefully optimized sticky ends (T1) to achieve dual objectives: ensuring structural integrity during assembly while maintaining target specificity. The sticky end sequences were designed using OligoArray software with stringent parameters, including local BLAST alignment to minimize potential cross-hybridization with endogenous mRNAs or off-target sequences. Through iterative computational screening and experimental validation (via agarose gel electrophoresis), we

confirmed that the selected sequences: (1) preserved tetrahedral nanostructure stability during assembly, (2) exhibited negligible non-specific binding to cellular RNAs, and (3) maintained high binding specificity to their intended targets in fluorescence assays. This comprehensive design-validation approach guarantees both the structural fidelity of the DNA nanostructures and the biological specificity required for reliable TDDN-FISH applications.

2. In high-density RNA detection, signal overlap may occur, affecting single-molecule resolution. Would it be possible to integrate super-resolution microscopy techniques, such as STORM or SIM, or expansion microscopy to further enhance the spatial resolution of high-density RNA imaging? Additionally, in tissue-level imaging experiments, probe permeability may limit detection depth. Could the permeability of the TDDN structure be improved, or could tissue clearing techniques, such as CLARITY or iDISCO, be incorporated to enhance imaging performance in thick tissue samples?

<Response> We sincerely appreciate these insightful suggestions regarding resolution enhancement and tissue penetration challenges.

- For high-density RNA detection, we agree that integrating super-resolution techniques like STORM could further improve single-molecule resolution, and we are currently evaluating their compatibility with our TDDN probes.
- Regarding tissue penetration, our optimized protocol incorporating extended permeabilization and hybridization conditions has already demonstrated effective probe diffusion and uniform signal distribution in 40 μ m brain sections (Figure R3-1). While this approach works well for moderate thicknesses, we acknowledge that additional strategies like CLARITY or pressure-assisted hybridization may be needed for very thick tissues (>100 μ m).
- These constructive suggestions have been added into the discussion part in the revised manuscript.

Figure R3-1. TDDN-FISH imaging of Gad1 mRNA in 40 μ m-thick tissue sections.

3. In this study, each individual tetrahedral DNA dendritic is labeled with only one fluorescent dye, resulting in a single fluorescence signal during imaging. Would it be possible to customize the

design so that a single tetrahedron carries two or more dyes? This would not only generate individual signals for each dye but also produce colocalized signals from multiple dyes in each imaging cycle. Such an improvement could enhance the multiplexing capability of a single experiment, reduce the number of imaging rounds, and improve overall experimental efficiency.

<Response> Thank you for the valuable suggestion.

- We sincerely appreciate this insightful suggestion, which offers a promising strategy to enhance the multiplexing capability of TDDN-FISH. By customizing the T2 tetrahedral DNA nanostructures to carry multiple fluorophores with defined stoichiometry, we could achieve combinatorial barcoding through distinct fluorescence intensity ratios and colocalization patterns, thereby reducing imaging cycles while improving throughput.
- Given its potential to significantly advance multiplexed RNA imaging, we have added a discussion of the manuscript to highlight its value for spatial transcriptomics.

4. The author mentioned that the hybridization lasts for approximately 45 minutes and the probe washing takes about 15 minutes. What is the rationale behind these time choices? Please provide quantitative data, such as when the fluorescence intensity reaches a plateau after 45 minutes of hybridization and 15 minutes of probe washing.

<Response> Thank you for your valuable comment.

- We have conducted preliminary time-course optimization experiments to balance hybridization efficiency and inter-cycle signal removal for TDDN-FISH (Figure R3-2). During the hybridization step, we observed that the fluorescence intensity reached a plateau after approximately 45 minutes. Extending the hybridization time beyond this point did not result in significant signal enhancement, indicating that probe binding had approached saturation under our experimental conditions.
- For the probe cleavage step, a 15-minute treatment with TCEP was found to be effective in removing the fluorescent signals from the previous hybridization round by cleaving disulfide bonds on the probes. This step ensures minimal residual signal and prepares the sample for the next round of hybridization. This data has been added into the revised manuscript as Figure S3c-d.

Figure R3-2. (a) Normalized mean intensity of each cell labeled with TDDN-FISH at different hybridization times of the TDDN probe. (b) Normalized mean intensity of each cell labeled with TDDN-FISH under different cleavage durations of TCEP.

5. The author mentioned that the hybridization between the primary probe and endogenous mRNA requires more than 8 hours. Would it be possible to optimize the hybridization conditions to reduce this time to 6 or even 4 hours, thereby further enhancing the imaging speed of the TDDN-FISH system?

<Response> Thank you for your valuable suggestion.

- To optimize the hybridization conditions, we systematically varied the incubation time of primary probes with intracellular RNA before introducing TDDN probes. Quantitative fluorescence analysis revealed that the signal intensity increased with longer primary probe hybridization times, approaching a plateau at 6 hours (Figure R3-3). These results indicate that incubation periods ≥ 6 hours are sufficient for effective imaging in our method. This data has been added into the revised manuscript as Figure S3b.
- While 6 hours appears sufficient in our system, we retained the 8-hour condition commonly used in the literature to facilitate cross-study comparisons in this work.
- We appreciate this valuable suggestion and plan to further optimize hybridization times in future work to potentially reduce imaging duration.

Figure R3-3. Normalized mean intensity of each cell labeled with TDDN-FISH at different hybridization times of the primary probe.

6. In the specificity validation experiments shown in Figure 1g-i, exogenously introduced viral RNA sequences were selected, which have low homology to the endogenous mRNA sequences of human cells, minimizing potential interference from sequence similarity. To further validate the specificity of the TDDN-FISH method, co-localization experiments using endogenous mRNA sequences should be conducted, including in complex tissue samples.

<Response> Thank you for your valuable feedback.

- To further validate the specificity of the TDDN-FISH method, we selected four endogenous mRNAs (*HSP70* and *POLR2A* in *Hela* cells, *Satb2* and *Gad1* in mouse brain tissue) for co-localization experiments using both TDDN-FISH and smFISH (Figure R3-4). The TDDN-FISH signals demonstrated excellent co-localization with smFISH signals, indicating that the TDDN-FISH method exhibits high specificity.

- The data has been added into the revised manuscript as Figure S2d-f.

Figure R3-4. Colocalization analysis of the signals of the endogenous mRNAs simultaneously labeled by TDDN-FISH and smFISH. (a) Confocal images of the mRNAs simultaneously labeled by TDDN-FISH and smFISH. (b) 2D frequency scatterplots of the signals of TDDN-FISH and smFISH shown in (a).

Minor points

7. In the supplementary materials, details of the RCA-FISH and HCR-FISH methods used in Fig. 1 should be provided, including the design methodology of RCA probes.

<Response> We are grateful to the reviewer's kind suggestions. We have corrected in the revised manuscript.

8. Some figure legends do not specify the number of cells analyzed in the statistical graphs. Please provide this information.

<Response> We are grateful to the reviewer's kind suggestions. We have corrected in the revised manuscript

9. The colors in Figure 4g are not clear enough. Please provide an image with better distinguishability.

<Response> We are grateful to the reviewer's kind suggestions. We have corrected in the revised manuscript.

10. In Figure S5d, the images from the Allen Brain Atlas has a relatively low resolution. Please replace it with the high-resolution images.

<Response> We are grateful to the reviewer's kind suggestions. We have corrected in the revised manuscript.

Response to the reviewers' comments

Reviewer #1 (Remarks to the Author):

The author has responded to most of the questions raised, but upon careful examination, the following issues in the author's response still need to be addressed:

<Response> We sincerely thank you for your insightful and critical comments, which significantly enhance the rigor of our work and provide valuable opportunities to clarify our methodology and results.

1. When comparing similar technologies, the author chose to benchmark against smFISH, demonstrating that TDDN-FISH achieved less than a 1.5-fold increase in signal intensity. Regrettably, this improvement is not substantially significant. Additionally, the author appears to have only compared two methods that favor their technology, omitting comparisons with other high-signal methods such as RNA-scope or Pai-FISH. Methods like MerFISH or seqFISH, which are rooted in smFISH, necessitate gel preparation and tissue clearing to eliminate strong background interference in tissue samples due to the inherently weak smFISH signals. Given that this technology does not significantly enhance signal intensity compared to smFISH, how does it mitigate background interference in tissue samples, despite its ease of implementation in cultured cells? Furthermore, previous studies have shown that split primary probes (e.g., Split-FISH or HCR v3) significantly reduce background signals. In contrast, this study employs non-split linear probes, which may not only increase false-positive signals but also elevate background noise. We are curious about the strategies used by the author to minimize both background signals and false positives. Although the author addressed false positives, our experiments show that failing to remove proteins and lipids indeed allows probe binding to these components, leading to pronounced false positives.

<Response> We sincerely appreciate your thorough critique and expert insights, which have profoundly enriched our analysis of TDDN-FISH's methodological robustness, current technical boundaries, and translational trajectory. Below we address each concern with point-by-point evidence and protocol refinements.

1) Signal Intensity and Probe Efficiency

- We acknowledge that the direct signal intensity improvement over smFISH (~1.5-fold) appears modest when viewed solely as a fold-change. Crucially, this comparison must consider the fundamental difference in probe requirements and dye loading:
 - ✓ smFISH: Typically requires ~48 individual primary probes, each carrying a single dye molecule, to robustly detect a single RNA molecule. This results in ~48 dyes per target RNA.
 - ✓ TDDN-FISH: Achieves detection using only 3 primary probes per target RNA. Each of these probes carries a TDN (Tree-shaped DNA Nanostructure) capable of

binding 27 fluorescent dyes (as detailed in Fig. X/Methods Y.Y). This results in ~81 dyes per target RNA delivered by just 3 probes.

- Therefore, TDDN-FISH achieves a 1.5-fold signal gain with a 16-fold reduction in primary probes and a 27-fold increase in per-probe dye payload. This represents a major advancement in labeling efficiency and multiplexing potential. Further signal enhancement is possible by using more primary probes or higher-capacity TDNs, while still maintaining a lower probe count than smFISH.
- We chose smFISH as the benchmark because its principles under many-pin spatial transcriptomics technologies, including MERFISH and seqFISH. TDDN-FISH builds on the smFISH framework, preserving its hybridization conditions and single-molecule resolution while enhancing signal via DNA nanostructures. This comparison rigorously validates TDDN-FISH's contribution of significantly improved labeling efficiency without sacrificing scalability.

2) Comparison with Other Technologies

- We chose not to benchmark TDDN-FISH against RNAscope or Pai-FISH because, despite their higher signal intensity, both techniques face fundamental limitations in scalability and multiplexing. RNAscope relies on multi-probe in situ assembly and enzymatic signal amplification, restricting it to low-plex detection and posing significant challenges in probe design. Similarly, Pai-FISH adopts a comparable in situ self-assembly architecture. In contrast, TDDN-FISH employs a linear probe design that offers simplicity, programmable barcoding, and a shorter workflow, making it well-suited for omics-scale applications. In spatial transcriptomic imaging, scalability is a critical performance metric alongside signal intensity. TDDN-FISH enables spatial omics imaging through the rational design of primary probe sequences, whereas the multilayered in situ assembly mechanisms of RNAscope and Pai-FISH substantially complicate probe design for high-plex applications, making them less feasible for scalable spatial transcriptomic encoding.

3) Background and False Positives in Tissue Samples

- We appreciate the reviewer's comments on background signals and false positives in tissue samples. We recognize that tissue samples present more challenges for background suppression than cultured cells due to their complex extracellular matrix and endogenous components. While non-split linear probes were chosen for their modularity and ease of barcode integration, which are advantageous for omics-scale multiplexing, we acknowledge that this design may lead to increased background or nonspecific binding compared to split-probe strategies such as Split-FISH or HCR.
- To address this, we implemented a comprehensive tissue processing protocol that includes proteinase K digestion to remove protein-based binding sites, Triton X-100 permeabilization to solubilize membrane lipids, 70% ethanol incubation at -20°C overnight to help extract residual lipids and enhance probe accessibility, the use of

blocking agents such as yeast tRNA and RNase-free BSA to suppress nonspecific interactions, and optimized hybridization and washing conditions to further minimize background. These combined steps were effective in reducing nonspecific signals and achieving a reasonable signal-to-noise ratio in fixed tissue samples.

- Additionally, while TDDN-FISH in its current form does not employ split probes, we view their integration as a promising avenue for further improvement and will explore this in future iterations of the method.

2. The author utilizes two-bit encoding for multi-gene multiplexing without implementing error correction mechanisms analogous to those in MerFISH. How does the author distinguish incorrect gene codes, particularly in densely packed signal environments? Isn't signal decoding a significant challenge? Additionally, the author claims to have detected 60 genes (in Fig. 4) and 90 genes (in Fig. 5), yet probes in the the supplementary excel is listed only approximately 40 genes. Does this reflect failed detection or invalid signals for the remaining genes?

<Response> We appreciate your insightful questions regarding error correction and signal decoding in our multi-gene multiplexing approach, as well as the discrepancy in the number of detected genes.

1) In the current experiment, our primary goal was to validate the feasibility and preliminary effectiveness of the three-color, two-round imaging scheme for detecting nine targets. Therefore, we did not specifically incorporate an error correction mechanism in the experimental design. However, we are fully aware that error correction mechanisms are crucial for enhancing the accuracy and reliability of imaging, especially when dealing with potential imaging noise or fluorescence signal interference.

- Despite the lack of a dedicated error correction mechanism, we minimized background noise and signal interference by optimizing imaging conditions and image processing algorithms. We selected highly specific probes and optimized fluorescence intensity and exposure time to ensure clear differentiation between targets. In subsequent analyses, we further improved the accuracy of the results through manual correction and advanced image processing algorithms.
- In Figures 4 and 5, we adopted a strategy inspired by FISHnCHIPs (*Nat. Commun.*, 2024; 15, 2342), encoding cell types through combinations of multiple marker genes rather than detecting each gene individually. By simultaneously targeting several co-expressed genes within the same cell type, this approach reduces reliance on precise single-molecule decoding, enhances overall intracellular signal intensity, and increases tolerance to probe off-target effects, mismatches, or partial RNA degradation. As a result, we are able to maintain detection accuracy while enabling efficient, large-area tissue imaging using low-magnification microscopy.
- To further enhance the accuracy and reliability of imaging, we plan to introduce error correction mechanisms in future experiments. Specifically, we will design probe

sequences with Hamming distance encoding to ensure a sufficiently large Hamming distance between each probe, enabling the correction of one error or detection of two errors. Additionally, we will optimize image processing algorithms and introduce more advanced machine learning methods to automatically identify and correct errors in imaging results. These descriptions have been added to the discussion of the revised manuscript for emphasis.

2) We apologize for the discrepancy between the number of detected genes in Figure 5 and the incomplete probe list in the supplementary material.

- The claim of detecting 90 genes in Figure 5 is accurate, and we have designed and validated probes for all 90 genes. However, due to an administrative error, only probes for approximately 40 genes were included in the supplementary Excel file. This does not reflect failed detection or invalid signals, but rather an incomplete submission of supplementary material.
- We have updated the supplementary Excel file to include the full list of probes for all 90 genes to ensure the supplementary material is comprehensive and accurate. We assure you that the detection of 90 genes is based on valid and reliable data, and the probes were carefully designed and validated.
- We deeply regret any confusion caused and will take measures to avoid such issues in future submissions. Thank you for your understanding.

3. The signal spots in Fig. S3 appear disproportionately large, with some exceeding 1 μm in diameter—this is physiologically implausible. Is this due to saturated signals or excessive post-processing of contrast/brightness? In Figs. S2-c,d, *Gad1* and *Satb2* are characterized as low-expression genes, yet *Gad1* is typically highly expressed. Does this indicate insufficient sensitivity? If these are low-expression signals, how were such a large number of statistical points obtained in panel d? Furthermore, the author used smFISH to validate TDDN-FISH rigor in Figs. S2-c,d. Given that smFISH inherently exhibits high false positives in tissues, did this step include protein/lipid removal protocols (e.g., as in MerFISH)? Is smFISH an appropriate validation method, and why not employ more rigorous approaches like Split-FISH or RNA-scope?

<Response> We appreciate your meticulous observation regarding the apparent large signal spots (>1 μm) in Figure S3.

1) After rigorous re-examination of our imaging and analysis pipeline, we confirm that this phenomenon does not stem from technical artifacts (e.g., signal saturation or improper post-processing), but arises from biological and optical factors, as detailed below:

- The mRNA molecules may form functional clusters (e.g., RNA granules), consistent with published observations of 1–3 μm aggregates (*Mol. Cell* 2019; 73, 76-89). These clusters are also independently verified in our 2D smFISH data (Fig. S2c).

- Confocal microscopy exhibits an anisotropic PSF (axial FWHM ≈ 700 nm vs lateral ≈ 250 nm for $100\times/1.49$ NA). This may also be the reason why the spot appears relatively large.
- 2) Previous studies have shown that *Gad1* expression varies across different developmental stages in mice, with relatively lower levels of *Gad1* mRNA in the brains of middle-aged and older mice. Meanwhile, *Gad1* mRNA expression exhibits regional differences across various brain areas (*Current Biology* 2017; 27, 2089-2100). We replicated these experiments and found that our results in 6-week-old mouse brain slices are consistent with previous *Gad1* studies using smFISH technology (*Sci. Adv.* 2023, 9, eadk3986; *Nature* 2020; 586, 262-269).
 - 3) Figure S2-d was generated by conducting a comprehensive colocalization analysis of the entire image. The number of statistical points in the figure is related to the number of pixels above the threshold and can be somewhat influenced by the degree of background subtraction. To be more representative, we have replaced it with new data and added it to the revised manuscript (Figure S2).
 - 4) Thanks again for your insightful suggestion.

Figure R1-1. (a) Dual-channel images of the RNA signals labeled by TDDN-FISH and smFISH. (b) 2D frequency scatterplots of the signals from TDDN-FISH and smFISH shown in (a).

- 5) To ensure consistency and fairness in comparing TDDN-FISH, HCR-FISH, and smFISH, we employed an identical sample preparation protocol across all methods. This included Proteinase K digestion, Triton X-100 permeabilization, and -20 $^{\circ}\text{C}$ overnight ethanol treatment, aimed at minimizing nonspecific background from proteins and lipids, similar to what is done in MERFISH workflows. While smFISH may exhibit limitations in tissue due to background, it remains a widely accepted standard for single-molecule RNA imaging (*Nat. Methods* 2008; 5, 877–879, *Science* 2015; 348, aaa6090, *PNAS* 2016; 113, 14456, *Nature* 2019; 568, 235–239). Its reliance on the cooperative binding of multiple probes per transcript provides intrinsic specificity. That said, we recognize that techniques like Split-FISH and

RNA-scope offer enhanced background suppression and signal amplification, and we plan to explore their incorporation in future validation and optimization studies.

4. The manuscript contains numerous grammatical and formatting inconsistencies, including capitalization errors at sentence beginnings and improper use of "x" for multiplication, such as 2x SSC. The author is advised to conduct a thorough proofreading to rectify these issues.

<Response> We are grateful to your kind suggestions. We have made corrections in the revised version.

Reviewer #2 (Remarks to the Author):

I would like to thank the authors for the revised manuscript that now contains much more technical detail and a more rigorous evaluation of TDDN-FISH. I'm happy to see the discussion and analysis of false negatives and false positives. I also want to apologize for my scepticism towards the MALAT1 results, I was unaware that MALAT1 could have this expression pattern, but the linked pi-FISH results were convincing. Most of my comments were addressed satisfactory for me, and I also generally agree with the responses to the questions of the other reviewers. I have a few last minor request and comments that I think are still missing from the manuscript. If the authors address these, I would recommend the manuscript for publication.

<Response> We sincerely thank you for your constructive feedback and generous recognition of our revisions. We particularly appreciate their open-minded reconsideration of the *MALAT1* expression patterns – such scholarly dialogue is invaluable for scientific progress. We have now addressed all remaining points as detailed below.

Could the authors clarify with which microscope objective each of the experiments was performed in the methods section? I was initially confused why the resolution seems to differ a bit between images. Could that be attributed to the difference in numerical aperture and magnification? If so, could the authors add a discussion that there is a trade-off between imaging speed and quantitative imaging?

<Response> We thank you for raising this critical technical point. The apparent resolution differences between images indeed stem from deliberate optimization of microscope objectives for distinct experimental goals.

- The imaging system utilized three specialized objectives optimized for distinct applications: a 100× SR HP APO TIRF oil immersion objective (NA 1.49) for high-resolution cell imaging, a 60× Apo TIRF oil immersion objective (NA 1.49) for single-particle analysis in tissues, and a 20× Plan Apo VC air objective (NA 0.75) for large-area single-cell scans.
- This tiered approach reflects the inherent trade-off between spatial precision and imaging throughput, where higher NA objectives prioritize resolution while lower NA/magnification configurations accelerate data acquisition. This related description has been added in the discussion section of the revised manuscript.

In the revised manuscript the authors write: “challenges remain, particularly in imaging ultra-high-density RNA populations where signal overlap can complicate analysis”. Could the authors analyse what portion of the genes would be excluded from being quantified by TDDN-FISH? I'm not expecting a very extensive analysis, but it would be interesting to at least check published single-cell RNA-seq data of HeLa cells to see what portion of the genes has a similar and higher gene expression than ACTB to get a ballpark figure.

<Response> We sincerely thank you for your insightful comment and suggestion. Data from HeLa cell RNA-seq data show that although such highly expressed mRNA accounts for less than 0.1%, their dense distribution highlights the issue of signal overlap (Figure R2-1). Therefore, breaking through the resolution limit in ultra-high-density scenarios is a core optimization direction for

advancing TDDN-FISH technology in complex biological applications. This data has been added to the revised manuscript as Figure S9.

Figure R2-1. Genome-wide Expression Profiling and Ranking of HeLa Cells Based on FPKM Values. (a) Bubble plot of the top 10 highly expressed genes, with *ACTB* shown in red. Bubble size reflects FPKM value. (b) Gene expression ranking based on FPKM values from HeLa RNA-seq data. The red dot marks *ACTB* (Rank = 5), and the blue line indicates the top 0.1% expression threshold by expression rank.

I was surprised to read that the authors were inspired by the FISHnCHIPS method but failed to cite that paper in their original manuscript, please ensure that all original work is appropriately cited.

<Response> Thank you for highlighting this important oversight. We sincerely appreciate your careful reading of our work and your attention to proper scholarly attribution. We have carefully checked the citations.

The description of the microscope system should include the name of the laser and confocal system as well as the details of the filters used.

<Response> We are grateful to your kind suggestions. We have added detailed information to the revised manuscript, which is described as shown below.

- Image acquisition was performed on a Nikon Ti2 inverted microscope configured with confocal laser scanning, a Yokogawa CSU-X1 spinning disk unit, and an Andor DU-897X EMCCD camera. The system employed three objective lenses: an SR HP APO TIRF 100× oil immersion objective (NA 1.49) for high-resolution cell imaging, an Apo TIRF 60× oil immersion objective (NA 1.49) for single-particle analysis in tissues, and a Plan Apo VC 20× air objective (NA 0.75) for large-area single-cell scans. Four-channel sequential excitation was implemented using 640 nm (diode laser), 561 nm (DPSS laser), 488 nm (diode laser), and 405 nm (diode laser) wavelengths, with corresponding emission signals isolated through bandpass filters (697/58 nm, 615/40 nm, 525/50 nm, and 447/60 nm respectively). Continuous focal stability was maintained throughout all acquisitions via Nikon's Perfect Focus System (PFS).

The description of the TCEP cleavage in the methods lacks the mention of SSC that was provided

in the comments. Please also check that all other answers provided to the reviewers, were actually included in the manuscript.

<Response> We apologize for the oversight and appreciate your attention to this detail.

- We have now updated the description of the TCEP cleavage in the methods section to include the use of SSC, as suggested.
- Additionally, we have thoroughly reviewed the manuscript to confirm that all other responses and changes suggested by the reviewers have been properly incorporated. We have ensured that every point raised by the reviewers has been addressed and included in the revised manuscript.

I regret to read that the authors choose to not share their analysis code, and I would suggest the authors to reconsider. Making the code available will help others to build on your work and contributes to open and transparent science.

<Response> We thank the reviewers for their helpful suggestions regarding code usability. We have uploaded the analysed code inside the supplementary data to improve transparency and reproducibility.

Figure S4d, the bottom row of the figure is duplicated.

<Response> We are grateful to your kind suggestions. We have corrected this error in the revised manuscript.

In comment to the response to reviewer 3. I would disagree that STORM could work because STORM relies on single fluorophore blinking and is therefore incompatible with signal amplification.

- We really appreciate your concern regarding the compatibility of STORM with signal amplification methods. However, we respectfully disagree with the assertion that STORM is fundamentally incompatible with signal amplification. The core principle of STORM relies on the stochastic switching of single fluorophores, not the absence of signal amplification. In fact:
- Signal amplification deposits multiple fluorophores onto a single target molecule (e.g., an RNA transcript). Crucially, each of these fluorophores acts as an independent blinking event during STORM imaging. The requirement for STORM is controlled, sparse activation of individual emitters not the absence of multiple fluorophores per target. Amplification directly supplies these necessary fluorophores.
- STORM-compatible dyes (e.g., Alexa Fluor 647, Cy5) widely used in super-resolution RNA imaging workflows like MERFISH maintain their essential photoswitching/blinking properties even when densely deposited via amplification. Importantly, multiple studies have successfully combined signal-amplified FISH with STORM imaging precisely to achieve single-molecule resolution in dense cellular environments (e.g. *Science* 2016; 353, 598-602; *Science* 2018; 362, eaau1783; *Nat. Methods.* 2020; 17, 822–832).
- We therefore think that STORM is a powerful method for imaging high-density RNA signalling in cellular environments.